# Distilling Out-of-Distribution Robustness from Vision-Language Foundation Models

Andy Zhou[1, 3], Jindong Wang[2], Yu-Xiong Wang[1], and Haohan Wang[1]

[1]University of Illinois at Urbana-Champaign
[2]Microsoft Research
[3]AI@UIUC

{andyz3, yxw, haohanw}@illinois.edu, jindong.wang@microsoft.com

## Abstract

We propose a conceptually simple and lightweight framework for improving the robustness of vision models through the combination of knowledge distillation and data augmentation. We address the conjecture that larger models do not make for better teachers by showing strong gains in out-of-distribution robustness when distilling from pretrained foundation models. Following this finding, we propose Discrete Adversarial Distillation (DAD), which leverages a robust teacher to generate adversarial examples and a VQGAN to discretize them, creating more informative samples than standard data augmentation techniques. We provide a theoretical framework for the use of a robust teacher in the knowledge distillation with data augmentation setting and demonstrate strong gains in out-of-distribution robustness and clean accuracy across different student architectures. Notably, our method adds minor computational overhead compared to similar techniques and can be easily combined with other data augmentations for further improvements.

## 1 Introduction

One of the goals of machine learning is to develop systems that can generalize effectively across diverse populations and environments, much like human intelligence. Despite the impressive advancements in neural networks that surpass human performance in various tasks in computer vision, their generalization capabilities remain inadequate when faced with out-of-distribution data, such as adversarial perturbations [60], unusual colors and textures [17, 58, 68], or challenging contexts [29].

One major line of research addresses this issue with more sophisticated training strategies [37], including adversarial training [39], data augmentation [18, 28, 79], or other regularizations [35, 46, 61, 73, 69]. In this paper, we focus on adversarial-training-based data augmentation [75], which can enhance the quantity and diversity of training data. In addition, theoretical work suggests achieving high robustness requires significantly more samples than clean accuracy [56]. This has also been shown empirically [3, 24], most recently with transformers [14] which achieve robustness on a variety of computer vision tasks. In addition to weak inductive bias, powerful model capacity, and grounding with language, these models are often trained with large-scale datasets [12, 14, 48, 76], up to billions of images [57] that encompass many real-world distribution shifts. As a result, these foundation models [1] exhibit remarkable zero-shot generalization, especially on natural distribution shifts such as artistic renderings, but require large amounts of compute and heavily parameterized models.

In this paper, we aim to connect these two lines of work. We investigate if it is possible to improve robustness by introducing a foundation model as a teacher to distill robust representations and help generate a diverse data augmentation. We conduct our analysis without requiring the teacher's large-scale dataset and focus on out-of-distribution robustness by introducing an image-to-image generative model to discretize optimized perturbations. We aim to leverage in-distribution data to a greater

extent and conduct our investigation with CLIP [48]. This marks a departure from existing work in knowledge distillation (KD), which tends to focus on smaller models and datasets. In fact, prior work [8] has called into question the utility of distilling from stronger teachers over training from scratch altogether. However, we find that although this *model capacity gap* can impair improvements in clean accuracy, distilling from robust teachers improves out-of-distribution robustness, even when leveraging only in-distribution data. Surprisingly, distilling on clean ImageNet images from CLIP with the original KD objective [32] results in a more robust ResNet34 than training with state-of-the-art regularization methods, despite a parameter difference of ~13.7x.

However, it is currently unclear in what settings a teacher's robustness can reliably transfer to a student and how to best combine distillation with data augmentation. We aim to answer this question both theoretically and empirically. We view adversarial training and data augmentation in the lens of domain generalization and prove that the diversity of the augmented samples leads to improved robustness. Our findings further suggest that foundation models make for strong teachers due to their diverse training distribution.

Building upon these findings, we introduce *discrete adversarial distillation (DAD)*, a KD framework that further distills the robustness of a teacher model by leveraging the adversarial examples of the *teacher* discretized by a VQGAN [15] as data augmentation. Notably, these samples are generated in an offline fashion, adding minor computational overhead compared to standard adversarial training. Intuitively, a foundation model will produce more diverse adversarial samples than a teacher trained on the same distribution, and we provide a theoretical framework using Wasserstein distance to formalize this proposition. Empirically, when distilling CLIP to a ViT-B, we achieve robust accuracy of 46.1% on ImageNet-Sketch [68] and 65.1% on ImageNet-Rendition [26], improving on the state of the art by 17.8% and 11.3% respectively. DAD can also be freely combined with existing regularization techniques, resulting in improvements in clean accuracy. In summary, our contributions are [1]

1. Establishing the KD for out-of-distribution robustness setting and a proposing a novel KD objective based on data augmentation

2. Providing a theoretical framework in KD for the choice of a teacher based on the diversity of the data augmentation

3. Proposing a novel data augmentation DAD that outperforms both adversarial training and distillation techniques on natural distribution shifts

## 2   Related Work

We define *out-of-distribution accuracy/robustness* as a model's performance on non-adversarial distribution shifts, *adversarial accuracy/robustness* to the case of robustness of adversarial examples, and *clean accuracy* as evaluation on a dataset drawn from the same distribution.

**Data augmentation.** Data augmentation is frequently used as regularization [75] by expanding the quantity and diversity of training data. This is often achieved through simple transformations such as rotations or image crops or more advanced techniques such as image mixing [79, 28], reinforcement learning [10, 81] or adversarial training [21, 31, 41] to find the optimal transformation.

Adversarial training (AT) was initially introduced to enhance model robustness by training with adversarial examples [39]. Although effective for defending against adversarial attacks, several works [64, 78, 70] have indicated a trade-off between adversarial clean accuracy in AT, limiting its effectiveness as a general data augmentation. Considerable efforts [47, 49] have been made to minimize this trade-off and directly use adversarial examples as data augmentation [50, 74], but there is still a considerable gap in out-of-distribution performance compared to foundation models like CLIP [48].

One line of work has recently been adapted to this issue. The model-based robustness paradigm [4, 52] leverages the disentangled latent representations of pretrained generative models to improve or validate out-of-distribution robustness [80], and can be used to improve adversarial examples. Most similar to our work is [23, 2, 41], which use a GAN or VAE [15, 34, 67] to discretize or discover adversarial examples. However, we leverage both a pretrained discretizer and foundation model, and adapt the AT objective to a knowledge distillation setting.

---

[1]code at https://github.com/andyz245/DiscreteAdversarialDistillation

**Knowledge distillation.** Knowledge Distillation (KD) is a technique for training a student model with guidance from a stronger teacher model, widely applied in vision and language tasks [6, 32, 55, 71]. Most works focus on improving the KD objective with different knowledge transfer objectives, such as feature distance [7, 54, 63], attention [77], distribution [45], activation boundary [30], and sample distance [38, 44, 66]. [8] raises the model capacity gap issue, where training suffers when the size of the student and teacher models differ, but we find that there is still benefit to distilling from a robust teacher. Another line of work, defensive distillation, aims to distill adversarial robustness from an adversarially trained teacher [20, 82, 83]. We have a similar goal, but for out-of-distribution robustness and propose a loss objective not previously explored in prior works.

## 3 Method

We consider a standard dataset $\{(x, y)\}_{n=1}^{N} \sim P^N$ where instances and their labels $(x_n, y_n)$ are drawn from a distribution $P$ and are used for training the student model $\theta$. We also consider a discretizer, $Q$, and a teacher model $\phi$ and we use $\phi(x)$ to denote the output of a model given the sample $x$. $a$ denotes a function that applies a data augmentation on $x$, also known as a transformation function. Additionally, $a \in A$, the class of all such functions.

### 3.1 Setup

Invariance is a desirable property where the model will have the same representation of an input after a transformation is applied. A model, $\theta$ is said to be invariant if $\theta(x) = \theta(x')$ for all $x \in U$, where $U$ is the set of all images that can be obtained by a transformation of $x$ by $a \in A$, which includes the identity transformation. $a$ ranges from worst-case imperceivable perturbations to real-world distribution shifts like artistic sketches [68]. In this paper, we focus on the latter, denoted as $\hat{a}$ and $\hat{A}$. We can consider $\hat{a}$ to represent an individual distribution $P$ and $\hat{A}$ to be drawn from the distribution of distributions $\hat{P}$. Our ultimate goal is to train $\theta$ to be invariant to transformations in $\hat{A}$. A model that maintains the same representation under transformations or distribution shifts of $x$ is said to be robust, which we define as the worst-case expected risk where

$$r_{\text{worst}}(P, \epsilon) = \max_{P':w(P',P) \le \epsilon} \mathbb{E}_{(x,y) \sim P'} \, l(\theta(x), y), \tag{1}$$

where $l$ is the loss function and $r$ depends on an anchor distribution $P$, and $\epsilon$, the maximum deviation allowed under the Wasserstein's distance metric. Similarly, we can define the expected robustness and expected risk in terms of an arbitrary distribution, including the training distribution.

$$r(P, \epsilon) = \mathbb{E}_{P' \sim \hat{P}:w(P',P) \le \epsilon} \mathbb{E}_{(x,y) \sim P'} l(\theta(x), y), \tag{2}$$

$$r(P) = \mathbb{E}_{(x,y) \sim P} \, l(\theta(x), y). \tag{3}$$

The robustness of the resulting model is highly dependent on $x'$, $P$, and the choice of data augmentation. It is also susceptible to adversarial attacks, where $x'$ is a worst-case perturbation of $x$. Adversarial robustness can be improved with adversarial training, which couples the outer minimization objective from (3) with an inner maximization objective in the following

$$\min \, \mathbb{E}_{(x,y) \sim P} \, [l(\theta(x), y) + \max \, l(\theta(x'), y)], \tag{4}$$

where $x' = x + \epsilon$, $\epsilon$ is the perturbation, and $l$ is the cross-entropy loss. This achieves adversarial robustness, but cannot generalize well to real-world domain shifts in $\hat{A}$. To address this, we consider a generative model, $Q$, trained on $P$ and can model $\hat{A}$. Passing an input through $Q$ in the maximization objective applies a worst-case transformation from $\hat{A}$. This modifies (4) to minimize the empirical semantic adversarial risk,

$$\min \, \mathbb{E}_{(x,y) \sim P} \, [l(\theta(x), y) + \max \, l(\theta(Q(x')), y)]. \tag{5}$$

### 3.2 Distillation from a robust teacher

Next, we consider a setting where we also have access to a pretrained $\phi$ invariant to distribution shifts in $\hat{A}$. This enables us to leverage knowledge distillation (KD). This modifies (3) in the following

$$\min \, \mathbb{E}_{(x,y) \sim P} [l_1(\theta(x), y) + l_2(\theta(x), \phi(x))], \tag{6}$$

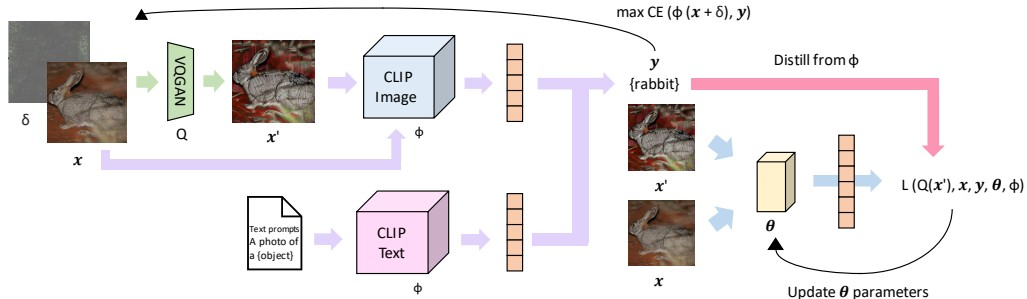

Figure 1: The overall framework of discrete adversarial distillation (DAD). We leverage a foundation model to generate and distill adversarial examples after discretization by a VQGAN.

where $l_1$ is the classification loss, the cross-entropy loss, and $l_2$ is a distance loss between $\theta(x)$ and $\phi(x)$, the KL divergence. (6) can be approximated by the empirical risk where

$$\min \ l_1(\theta(x), y) + l_2(\theta(x), \phi(x)). \tag{7}$$

Following theoretical work [43], distilling from a robust teacher with $l_2$ improves generalization due to minimizing the population risk, which has lower variance. In this formulation, the output of the teacher acts as a more robust supervisory signal than the label, encompassing the probability distribution of classes. This allows the student to learn the teacher representations on in-distribution data, but our experiments show that this is inadequate for out-of-distribution robustness, even when using a robust teacher. To address this, we combine (5) and (7) to also distill the representations of $\phi$ on augmented samples,

$$\min \ l_1(\theta(x), y) + l_2(\theta(x), \phi(x)) + l_2(\theta(Q(x')), \phi(Q(x'))). \tag{8}$$

The teacher is more robust, and is able to "solve" the perturbation for the student through distillation. Like [41] and [72], we train our models with both the original and augmented samples, expanding the size of the dataset and maintaining the original information path for $x$. We do not use the cross-entropy loss or $y$ labels for $x'$, as these labels may be wrong and could limit the expressiveness of the data augmentation. This allows us to use adversarial samples of the teacher in a novel maximization objective and obtain stronger empirical results.

### 3.3 Discrete Adversarial Distillation

The goal of our method, discrete adversarial distillation (DAD), is to distill from a large-scale vision-language model using only ImageNet [13] data. In the practical setting, we use approximations of an ideal discretizer and robust teacher. For $\phi$ we use CLIP [48], which was trained on a large-scale dataset and achieves impressive zero-shot generalization on a variety of natural distribution shifts.

For $Q$, we use a pretrained VQGAN [15], following [41], which also finds minimal improvements with a stronger discretizer. The VQGAN consists of an encoder, decoder, and quantization, where the encoder learns a latent representation of the input $x$, the quantization maps the representation to a visual codebook entry, and the decoder reconstructs $x$. We denote this process as $Q(x)$. In the adversarial training setting, $Q$ discretizes $x$, a worst-case perturbation $\epsilon$ is added by approximating the maximization objective to obtain $x'$, and the resulting image is then discretized by $Q$ again. To improve the transfer of robustness from the teacher, we hypothesize a better augmentation is more informative and exposes more teacher knowledge. We make this rigorous in the following section.

To generate these examples, we adopt adversarial training and modify the maximization objective of (5) to use the worst-case transformations of the teacher. We hypothesize a teacher trained on more diverse data will have more informative adversarial examples. To ensure the correctness of the perturbation, we only use samples that are still classified correctly by the teacher after generation. We use the teacher as an "oracle", allowing it to distill correct representations of the transformed image. Additionally, we generate these samples in an offline manner asynchronously from the pretrained teacher and add them to the original dataset during the training of the student. These examples

only need to be generated once for each teacher and be reused as additional data. This adds minor additional training costs compared to online adversarial training or DAT [41], which has 11x and 3.5x the cost of standard training, respectively [41]. Our full objective is described as,

$$\min \; l_1(\theta(x), y) + l_2(\theta(x), \phi(x)) + l_2(\theta(Q(x')), \phi(Q(x'))). \tag{9}$$

$$\text{where } x' = \max_{||x'-x||_p \leq \epsilon} l_1(\phi(x), y), \; \phi(Q(x')) = y.$$

### 3.4 Theoretical Investigation

We aim to investigate how to best distill robustness from a robust teacher trained on a large-scale dataset. We find that robust performance can be connected to the distance between the training and test distributions. A data augmentation can represent a new distribution, and the robustness of a model trained on this distribution can be quantified by its diversity and closeness to the test distributions. Although its representation on in-distribution samples can distill a degree of robustness, we show that due to being closer to the test distribution, it is more effective to leverage the discretized adversarial examples of the teacher than the student as our data augmentation of choice.

We begin with some assumptions.

**Assumption 1.** *For an arbitrary data pair $(x, y)$, transformations in $\hat{A}$ do not alter the semantics of the data. We can also say we consider an ideal labeling function where any $(x, y)$ pair can be correctly mapped, $y = f(x)$*

**Assumption 2.** *Any arbitrary distributions $P$ and $P'$ we compare possess smooth probability densities controlled by two constants $c$ and $\alpha$ depending on the smoothness of $P$ and $P'$ where $cw(P, P')^\alpha$ and $c > 0$ and is only dependent on $\alpha$.*

**Assumption 3.** *For function $\gamma(|\Theta|, n, \beta)$ parameterized by hypothesis space $|\Theta|$, number of samples $n$, and the probability when the bound holds $\beta$, if the samples are i.i.d, $\gamma(|\Theta|, n, \beta) = 2\mathcal{R}(\mathcal{L}) + \sqrt{(\log 1/\beta)/2n}$, where $\mathcal{R}(\mathcal{L})$ stands for Rademacher complexity and $\mathcal{L} = \{l_\theta \,|\, \theta \in \Theta\}$, where $l_\theta$ is the loss function corresponding to $\theta$. Additionally, if $\Theta$ is finite, $l(\cdot, \cdot)$ is a zero-one loss, and samples are i.i.d, then $\gamma(|\Theta|, n, \beta) = \sqrt{(\log(|\Theta|) + \log(1/\beta))/2n}$*

**Lemma 3.1.** *Given Assumptions 1 and 2 and variational divergence $tv$, for two arbitrary distributions $P$ and $P'$ with corresponding density functions $\delta$ and $\delta'$, $r(P') \leq r(P) + w(P', P)$.*

**Lemma 3.2.** *Given Assumption 3, Lemma 3.1, and probability at least $1 - \Gamma$,*

$$r(P') \leq r((X, Y)_P) + w(P', P) + \xi(n_{(X,Y)_P}, \Theta, \Gamma)$$

*where $n_{(X,Y)_P}$ denotes the number of sample sizes in the finite dataset $(X, Y)_P$, and $\xi$ is a vanilla term that connects $n_{(X,Y)_P}$ and $\Theta$ with the generalization error bound.*

*Proof.* We apply conventional generalization analysis through uniform convergence to Lemma 3.1. We leave the full proof of Lemma 3.1 in Sec. E in the Appendix. $\qquad\square$

This results in an intuitive conclusion: empirical robustness performances depends on the divergence between the training and testing distributions, as well as two additional elements. The first is the empirical error term on the training distribution, which can be quantified, and the second is a technical term influenced by the sample size and hypothesis space. The exact manner in which $\xi$ depends on these parameters is contingent on the particular convergence analysis employed.

Therefore, the decisive factor for robustness is the degree of deviation between the training and testing distributions. Therefore, using diverse data augmentations close to the testing distribution will lead to the largest gains in robustness. This intuitive understanding suggests that training with distributions generated from the teacher will be more advantageous, as the teacher, having been trained on a large dataset, encapsulates more diverse distributions.

There findings are applicable to any arbitrary distributions $P \sim \hat{P}$. Nevertheless, this doesn't inherently encapsulate the characteristics of foundation models that trained on data from across the internet, composed of a variety of semantic distributions from $\hat{A}$.

Therefore, we use $\mathbf{P} \in \hat{A}$ to denote a set of $m$ distributions, i.e., $\mathbf{P} = \{P_1, P_2, \ldots, P_m\}$, and we consider a pretrained foundation model trained with such a set of distributions. To facilitate forthcoming discussions, we extend the notation of $w$ to encompass sets, defining $w(P', \mathbf{P})$ as the average divergence between distributions within the set. Thus, $w(P', \mathbf{P}) := \sum_i^m w(P', P_i)/m, \quad \forall P_i \in \mathbf{P}$.

**Lemma 3.3.** *Given a distribution $P$, we generate a new distribution $P^* \in \hat{A}$ using the discretized worst-case adversarial samples of a model $\theta$. Training $\theta$ with adversarial training is equivalent to training $\theta$ with empirical risk minimzation on $P^*$ where $w(P, P^*) \leq \epsilon$.*

We leave the full proof of Lemma 3.3 in Sec. E of the Appendix. Finally, let's denote the model $\phi$ trained over distribution $P$ as $\phi(P)$ and the adversarial data augmentation process that results in a new semantic data distribution as $D$. We aim to compare $w(D(\phi(\mathbf{P})), P')$ and $w(D(\phi(P)), P')$. In this context, $P'$ is any arbitrary testing distribution, $P$ is a specific training distribution, and $\mathbf{P}$ represents the set of distributions used for training foundation models.

**Lemma 3.4.** *Assuming $\hat{P}$ is continuous and has a finite expected value, for two sets of distributions $\mathbf{P}_1$ and $\mathbf{P}_2$, assuming there is at least one distribution in the intersection of $\mathbf{P}_1$ and $\mathbf{P}_2$, for a fixed testing distribution $P'$, we have*

$$\mathbb{E}_{\hat{P}}\Big[\big|w(D(\phi(\mathbf{P}_1)), P') - w(D(\phi(\mathbf{P}_2)), P')\big|\Big] \leq 2\mathbb{E}_{\hat{P}}\Big[\sup_{P \in \mathbf{P}_1} w(P, P') + \sup_{P \in \mathbf{P}_2} w(P, P')\Big]$$

We leave the full proof of Lemma 3.4 in Sec. E of the Appendix. Our findings provide a comparison of the differences in the training mechanisms for various models $\phi$, each trained with distinct data sets ($\mathbf{P}_1$ and $\mathbf{P}_2$) and subjected to adversarial training. The methodology can easily be simplified to compare the bounded results between adversarial training based on the teacher model and standalone adversarial training by setting one of the training datasets to consist of a single distribution.

In the scope of our investigation, we compare DAD and discrete adversarial training based on the teacher model, referred to as $\mathbf{P}_1$, with DAT [41] and the traditional approach of adversarial training based on the student model, referred to as $\mathbf{P}_2$. Our results suggest two key implications:

1. Given a fixed $\mathbf{P}_2$, a more diverse $\mathbf{P}_1$ potentially results in greater variations in performance. We show visualizations that support this in Sec. D in the Appendix. In other words, the use of larger, more diverse pretrained datasets for the teacher model or more diverse data augmentations for the student model increases the likelihood of creating a robust final model. This is also been shown empiricially in prior work investigating the source of robustness in foundation models [16, 62]. However, in practice, the efficacy of distilling from this teacher depends on a variety of factors, including student-teacher architecture, training objective, and student capacity.

2. For a fixed $\mathbf{P}_2$, the greater the distance between the testing dataset $P'$ and $\mathbf{P}_2$, the more likely it is that the performance gains will be realized by distilling the teacher model trained on a more extensive set of training data. To put it intuitively, if the testing dataset closely resembles the training dataset (i.e., $w(\mathbf{P}, P')$ is small), then adversarial training based on the teacher model might not yield significant performance improvements. However, if the testing dataset differs considerably from the training dataset, then adversarial training based on a teacher model that has been trained on a larger dataset is more likely to yield superior performance gains. This observation aligns with our empirical results.

## 4 Experimental Results

### 4.1 Experimental Setup

**Models.** We focus primarily on ResNet50 [25] and ViT-B/16 [14]. We distill from a frozen pretrained CLIP-ViT-L/14 [48], trained on 224 x 224 resolution images with a patch size of 14.

**Datasets.** We train our models on ImageNet-1K [13]. We use several evaluation datasets. For in-distribution performance, we evaluate on ImageNet and ImageNet-V2 [51], a replication of ImageNet's evaluation set. We focus our study on natural distribution shifts and evaluate on ImageNet-A [29], a set of adversarially filtered natural images misclassified by a ResNet50, ImageNet-Sketch [68] which contains artistic sketches of objects, and ImageNet-Rendition [26] which contains abstract

Table 1: Main results on natural distribution shifts and in-distribution ImageNet. Baseline models are ViT-B/16 (top half) and ResNet50 (bottom half) trained on 224 x 224 images. The CLIP teacher is ViT-L/14. DAD variants have the best average performance for both types of distributions.

| Method | Rendition | Sketch | A | Avg | Method | ImageNet | V2 | Avg |
|---|---|---|---|---|---|---|---|---|
| CLIP [48] | 87.7 | 61.6 | 64.2 | 71.2 | CLIP [48] | 79.9 | 72.9 | 76.4 |
| ViT [14] | 27.1 | 17.3 | 8.0 | 17.5 | ViT [14] | 72.8 | 58.7 | 65.8 |
| Advprop [74] | 43.5 | 31.7 | 18.5 | 31.2 | Advprop [74] | 79.5 | 68.7 | 74.1 |
| Fast Advprop [42] | 41.8 | 29.4 | 17.9 | 29.7 | Fast Advprop [42] | 79.0 | 67.0 | 73.0 |
| Debiased [36] | 40.3 | 29.4 | 18.3 | 29.3 | Debiased [36] | 79.3 | 67.6 | 73.5 |
| AugReg-ViT [59] | 39.5 | 29.2 | 19.0 | 29.2 | AugReg-ViT [59] | 79.9 | 67.9 | 73.9 |
| + Pyramid AT [31] | 47.7 | 36.8 | 23.0 | 35.8 | + Pyramid AT [31] | 81.7 | 70.3 | 76.0 |
| + DAT [41] | 47.3 | 34.8 | 30.2 | 37.4 | + DAT [41] | 81.5 | 70.8 | 76.2 |
| **+ DAD (Ours)** | **65.1** | **46.1** | **31.8** | **47.7** | **+ DAD (Ours)** | 79.6 | 69.9 | 74.8 |
| **+ DAT + DAD (Ours)** | 53.2 | 39.3 | 28.2 | 40.2 | **+ DAT + DAD (Ours)** | **81.9** | **71.7** | **76.8** |
| ResNet50 [25] | 36.1 | 24.0 | 0.0 | 20.0 | ResNet50 [25] | 76.1 | 63.2 | 69.7 |
| Advprop [74] | 38.8 | 25.5 | 4.3 | 22.9 | Advprop [74] | 77.6 | 65.5 | 35.6 |
| Pyramid AT [31] | 38.9 | 23.8 | 3.0 | 21.9 | Pyramid AT [31] | 75.5 | 62.5 | 71.6 |
| Debiased [36] | 40.8 | 28.4 | 3.5 | 24.2 | Debiased [36] | 76.9 | 65.0 | 71.0 |
| DAT [41] | 42.0 | 27.3 | 4.4 | 24.6 | DAT [41] | 76.5 | 65.0 | 70.8 |
| **DAD (Ours)** | **51.6** | **35.8** | **7.7** | **31.7** | **DAD (Ours)** | 75.7 | 65.0 | 70.4 |
| **DAT + DAD (Ours)** | 47.7 | 33.3 | 6.1 | 29.0 | **DAT + DAD (Ours)** | **77.8** | **66.0** | **71.9** |

or rendered objects. To observe performance on distributions that are out-of-distribution for the CLIP teacher, we evaluate on synthetic distribution shifts ImageNet-C [27] , which applies corruptions (snow, blur, noise, etc.) to ImageNet, and Stylized-ImageNet [17], which processes ImageNet with style transfer from a source image.

## 4.2 Baselines

DAD consists of both a data augmentation and knowledge distillation objective. We compare to both types of methods in our experiments.

**Common data augmentations.** For the simplest baseline, we follow [59] and train with common data augmentations Mixup [79], which combines images and labels, and Randaugment [11], which learns a policy over common transformations such as brightness or shear.

**DAT.** We also compare against the state-of-the-art data augmentation, DAT [41], which uses a VQGAN [15] to discretize adversarial examples in adversarial training. DAT uses the standard adversarial training objective 5.

**Knowledge distillation.** We compare against other logit-based knowledge distillation objectives, which only distill the output logits of the teacher. We consider standard knowledge distillation 7 and DIST [33], which aims to address the model capacity gap issue by distilling logit class relationships. Neither method natively supports distillation on augmented samples, so we also compare to defensive distillation objectives ARD [20] and RSLAD [83]. ARD modifies 7 to use the KL divergence between the student logits on the augmented sample with the teacher logits on the normal sample. RSLAD is an extension of ARD that replaces the cross-entropy terms with a KL divergence loss. For a fair comparison, we use DAD as the data augmentation.

## 4.3 Main Experimental Results on ViT-B/16 and ResNet50

**ImageNet-1K.** Tab. 1 shows results on ImageNet-1K and its distribution shifts. We compare against ViT-B/16 and ResNet50 models without data augmentation and with the state-of-the-art data augmentation approaches, PyramidAT [31] and DAT [41]. We combine DAD with the data augmentations used in AugReg [59], MixUp [79] and RandAugment [11]. We find that DAD has the best average performance across datasets for both models. For ViT-B we find that DAD has competitive in-distribution performance, but greatly improves performance on natural distributions. Compared to Pyramid AT and DAT, DAD also generalizes well to ResNet50. This suggests that the

Table 2: Main results on synthetic distribution shifts, which is out-of-distribution for the CLIP teacher. Models are ViT-B/16 (left) and ResNet50 (right) trained on 224 x 224 images. The CLIP teacher is ViT-L/14. Compared to DAT, DAD tends to perform worse on ViT-B but better on ResNet50.

| Method | C (↓) | Stylized |
|---|---|---|
| CLIP [48] | 60.2 | 18.5 |
| ViT [14] | 74.0 | 6.4 |
| Advprop [74] | 51.5 | 19.2 |
| Fast Advprop [42] | 53.3 | 18.4 |
| Debiased [36] | 49.8 | 22.4 |
| AugReg-ViT [59] | 54.5 | 16.6 |
| + Pyramid AT [31] | 45.0 | 19.1 |
| + DAT [41] | **44.7** | **23.1** |
| **+ DAD** | 53.2 | 22.4 |
| **+ DAT + DAD** | 47.5 | 22.6 |

| Method | C (↓) | Stylized |
|---|---|---|
| ResNet50 [25] | 76.7 | 7.4 |
| Advprop [74] | 70.5 | 8.0 |
| Pyramid AT [31] | 76.4 | 10.4 |
| Debiased [36] | 67.6 | **17.4** |
| DAT [41] | 74.2 | 10.8 |
| **DAD** | 67.4 | 13.1 |
| **DAT + DAD** | **65.2** | 14.6 |

Table 3: Comparison to distillation objectives on ImageNet. We use DAD for the data augmentation of ARD and RSLAD for a fair comparison. All students are ViT-B/16 trained on 224 x 224 images and all teachers are CLIP-ViT-L/14. We find that our distillation objective is the best at distilling out-of-distribution robustness from CLIP.

| | In-distribution | | Synthetic | | Natural | | | |
|---|---|---|---|---|---|---|---|---|
| Method | ImageNet | V2 | C (↓) | Stylized | Rendition | Sketch | A | Avg |
| KD [32] | 78.6 | 67.2 | 61.5 | 16.2 | 51.5 | 34.7 | 16.0 | 43.2 |
| DIST [33] | 76.6 | 63.9 | 65.8 | 12.7 | 40.8 | 26.9 | 11.2 | 38.0 |
| ARD [20] | **80.1** | **70.3** | **52.1** | 22.2 | 55.6 | 38.6 | 27.3 | 48.9 |
| RSLAD [83] | 79.9 | 69.3 | 55.6 | 20.8 | 55.9 | 39.8 | 25.5 | 47.9 |
| DAD (Ours) | 79.6 | 69.9 | 53.2 | **22.4** | **65.1** | **46.1** | **31.8** | **51.7** |

DAD data augmentation can be used across student architectures and that due to distillation, DAD is especially effective when training smaller models.

We also demonstrate DAD can be combined with existing approach DAT for stronger in-distribution performance. We add our distillation objective to the DAT objective and train the student on both the teacher's and its own adversarial samples. However, this comes at the cost of lower performance on natural distribution shifts, although we do observe that DAD + DAT still outperforms the prior state-of-the-art on ImageNet-Sketch and ImageNet-Rendition.

**Synthetic distribution shifts.** We also evaluate our models on synthetic distribution shifts composed of generated transformations in Tab. 2. Since the diverse training distribution of CLIP is mostly composed of natural distribution shifts, it has weaker zero-shot generalization to synthetic distribution shifts, and this performance is inherited in the student model. In fact, zero-shot CLIP is already outperformed by some compared methods on ImageNet-C, and Stylized-ImageNet. However, for ResNet50 DAD also has the best ImageNet-C performance, likely due to compared methods being specialized for certain distribution shifts [36, 74] or architectures [31].

**Distillation.** In Tab. 3 we compare DAD to knowledge distillation objectives. KD [32] and DIST [33] are vanilla distillation approaches without data augmentation or AT. ARD [20] and RSLAD [83] are defensive distillation objectives that use a adversarially robust teacher to encourage invariance to perturbations. For a fair comparison, we use CLIP-ViT-L/14 as the teacher and discretize the adversarial examples. We find that our distillation objective outperforms vanilla and defensive distillation objectives. We note that all methods can transfer robustness to the student, even methods without data augmentation.

**ImageNet-21k.** In Tab. 4 we show further gains in robustness from applying DAD to a ViT-B/16 pretrained on ImageNet-21K. We fine-tune this model with our method using only our method. Despite the baseline model performing worse than the variant trained with AugReg, DAD achieves

Table 4: Main results from pre-training on ImageNet-21K and fine-tuning on ImageNet-1K. All columns report top-1 accuracy except ImageNet-C which reports mean Corruption Error (mCE) where lower is better. All models are ViT-B/16 trained on 224 x 224 images. We find that pretraining on ImageNet-21K results in larger robustness improvements.

| Method | In-distribution | | Synthetic | | Natural | | | |
| | ImageNet | V2 | C (↓) | Stylized | Rendition | Sketch | A | Avg |
|---|---|---|---|---|---|---|---|---|
| ViT | 77.5 | 65.7 | 61.9 | 17.7 | 41.5 | 16.4 | 23.1 | 40.0 |
| DAT [41] | **83.1** | **73.2** | **43.6** | **24.8** | 55.2 | 41.7 | 36.7 | 53.0 |
| DAD (Ours) | 79.8 | 70.9 | 52.0 | 23.4 | **72.1** | **51.2** | **40.3** | **55.1** |

Table 5: Results on other student/teacher architectures, on ImageNet-1K. All experiments use DAD for the knowledge distillation objective and data augmentation. Using the best CLIP model as the teacher tends to result in the highest overall performance, but some teachers are better for some shifts.

| Student | Teacher | IM | A | C (↓) | V2 | Rendition | Sketch | Stylized | Avg |
|---|---|---|---|---|---|---|---|---|---|
| ViT-B | CLIP-RN101 [48] | **81.2** | 24.3 | **49.6** | 69.3 | 48.7 | 34.5 | 17.3 | 46.5 |
| ViT-B | CLIP-ViT-L [48] | 79.6 | **31.8** | 53.2 | **69.9** | **65.1** | **46.1** | **22.4** | **51.7** |
| RN50 | - | 76.1 | 0 | 76.7 | 63.2 | 36.1 | 24.0 | 7.4 | 32.9 |
| RN50 | ViT-B + DAT [41] | **80.4** | **10.1** | **65.6** | **68.8** | 40.4 | 29.6 | 8.5 | 38.9 |
| RN50 | DrViT-S [40] | 78.5 | 5.5 | 67.4 | 66.2 | 42.0 | 30.1 | 11.5 | 38.1 |
| RN50 | CLIP-RN101 [48] | 76.4 | 5.4 | 70.2 | 64.5 | 47.7 | 32.2 | 9.6 | 37.9 |
| RN50 | CLIP-ViT-L [48] | 75.7 | 7.7 | 67.4 | 65.0 | **51.6** | **35.8** | **13.1** | **40.2** |
| ViT-S | - | 77.8 | 11.9 | 63.9 | **66.0** | 36.9 | 25.3 | 12.0 | 38.0 |
| ViT-S | ViT-B + DAT [41] | **77.8** | 11.9 | 67.1 | 66.0 | 36.9 | 25.3 | 12.0 | 37.5 |
| ViT-S | CLIP-RN101 [48] | 73.4 | 9.0 | 65.2 | 62.1 | 38.8 | 23.9 | 12.0 | 36.3 |
| ViT-S | CLIP-ViT-L [48] | 73.8 | **18.0** | 63.1 | 64.0 | **52.9** | **35.8** | **17.3** | **42.7** |
| RN34 | - | 66.5 | 3.0 | 94.5 | 54.7 | 32.4 | 21.0 | 5.6 | 27.0 |
| RN34 | RN50 + AugMix [28] | 68.9 | 1.8 | 82.9 | 56.2 | 37.2 | 24.1 | 9.9 | 30.7 |
| RN34 | DrViT-S [40] | 68.2 | 2.1 | 79.5 | 55.6 | 37.0 | 23.0 | 10.5 | 31.0 |
| RN34 | ViT-B + DAT [41] | **69.2** | 2.2 | **79.0** | **56.6** | 38.7 | 25.0 | 11.0 | 32.0 |
| RN34 | CLIP-RN101 [48] | 65.4 | 2.5 | 85.3 | 53.5 | 42.5 | 26.3 | 8.5 | 30.5 |
| RN34 | CLIP-ViT-L [48] | 63.6 | **4.5** | 82.0 | 53.7 | **46.0** | **29.1** | **11.7** | **32.4** |

higher relative and absolute gains in robustness. We hypothesize the larger training distribution equips the student with useful inductive biases that let it more easily learn the more out-of-distribution adversarial examples generated from CLIP. We note that the CLIP training set is still ∼28.2x larger so this does not contradict our theory, but it may also be beneficial to train or pretrain the student on a more diverse dataset for a smoother distillation process.

## 4.4 Ablations

**Other student and teacher architectures.** Although our method and theory is adapted for foundation models, we investigate its efficacy on other models and teachers in Tab. 5. We consider a different large-scale teacher, CLIP-RN101, as well as teachers trained on ImageNet-1K that achieve out-of-distribution robustness through methods besides large-scale training, like Discrete ViT [40] or ViT-B [14] trained with DAT [41]. We also consider smaller students like ResNet34 [25] and ViT-S.

We find that distilling robustness in our setting depends on several factors, but most crucially, the robustness of the teacher. Like other distillation techniques, we find that our method can transfer representations between various student/teacher architectures. We find that our method is also susceptible to the model capacity gap, with lower clean accuracy on ResNet34 when distilling from CLIP than training from scratch. However, using CLIP results in the best performance on natural distribution shifts. Despite the more similar architecture, distilling from CLIP-RN101 across students is less effective than distilling from the more robust CLIP-ViT-L. We include similar results with vanilla knowledge distillation in Sec. A of the Appendix.

Table 6: Comparisons to data augmentation approaches. All columns report top-1 accuracy except ImageNet-C which reports mean Corruption Error (mCE) where lower is better. All models are ViT-B/16 trained on 224 x 224 images. We remove the distillation terms and use DAD samples as a standard data augmentation. All methods are based on AugReg and use Mixup and Randaugment.

| Model | IM | A | C ($\downarrow$) | V2 | Rendition | Sketch | Stylized | Avg |
|---|---|---|---|---|---|---|---|---|
| AugReg-ViT [59] | 79.9 | 19.0 | 54.5 | 67.9 | 39.5 | 29.2 | 16.6 | 42.5 |
| + Pyramid AT [31] | **81.7** | 23.0 | 45.0 | 70.3 | 47.7 | 36.8 | 19.1 | 47.7 |
| + DAT [41] | 81.5 | **30.2** | **44.7** | **70.8** | 47.3 | 34.8 | **23.1** | **49.0** |
| **+ DAD (Ours)** | 80.2 | 24.6 | 53.0 | 69.8 | **51.7** | **36.9** | 22.1 | 47.5 |

Table 7: Adversarial-training-based data augmentations and their training budget. Cost is based on training a ResNet50 from scratch for 100 epochs. While still more expensive than standard training, DAD is significantly cheaper than other techniques due to reusing precomputed adversarial examples.

| Method | Attack Steps | Training Budget |
|---|---|---|
| ImageNet | 0 | 1x |
| Adversarial Training [22] | 10 | 11x |
| AdvProp [74] | 5 | 7x |
| Fast AdvProp [42] | 1 | 3x |
| DAT [41] | 1 | 3.5x |
| **DAD (Ours)** | 1 | 2x |

**Pure data augmentation.** We study in Tab. 6 the effect of training on the DAD adversarial examples purely as a data augmentation technique, without distillation. Although DAD remains competitive, we find significant drops in performance, suggesting that it is difficult for the student to learn robust representations of these images on its own. However, we continue to observe improvements on natural distribution shifts, suggesting these samples are closer to CLIP's training distribution. However, training with DAD samples is significantly cheaper than DAT and Pyramid AT, making it more efficient in practice.

**Computational cost analysis.** Since DAD uses adversarial examples generated from a frozen teacher, there is no need to regenerate them during training. This amortizes the otherwise significant cost of adversarial training. We compare the cost of DAD with other adversarial data augmentation approaches in Tab. 7. By avoiding the need to continuously generate new adversarial examples, the only remaining cost for DAD is training on a larger dataset, making it cheaper than similar methods.

Additional ablations on choice of generative model, use of gradients, and transfer to adversarial robustness can be found in Sec. B in the Appendix.

## 5 Conclusion and limitations

We conduct the first study on distilling out-of-distribution robustness. We develop a framework for the use of foundation models in this setting and empirically and theoretically validate their advantages as a teacher. We propose discrete adversarial distillation (DAD) which uses the discrete adversarial examples of the teacher as a more diverse data augmentation and directly distill its most diverse representations. However, we find that DAD tends to be biased towards the performance of the CLIP teacher, exhibiting improvements mostly on natural distribution shifts. In practice, these shifts tend to be the most useful, and with the small computational cost of using DAD, we encourage practitioners to adopt it when training small models. We hope the development and release of improved foundation models and generative models will further demonstrate the effectiveness of our method.

We encourage further work to understand the limitations of machine vision models in out-of-distribution settings. More robust models carry the potential risk of automation bias, i.e., an undue trust in vision models. However, even if models are robust against corruptions in finite out-of-distribution datasets, they might still quickly fail on the massive space of semantic transformations in real-world data. Understanding under what conditions model decisions can be deemed reliable is still an open research question.

## Acknowledgements

This work was supported in part by NSF Grant 2106825 and NIFA Award 2020-67021-32799.

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

# 6 Appendix

The appendix is organized as follows. First, in Sec. A, we show additional results on using the original knowledge distillation objective. In Sec. B we show the results of additional ablations on the generative model, use of gradients, and transfer to adversarial robustness. In Sec. C we provide additional hyperparameter and implementation details. In Sec. D we show visualizations of DAD and the diversity of its data augmentation to support our theory. In Sec. E we provide full proofs from our theory. Finally, in Sec. F we provide visualizations of generated DAD samples compared to standard and DAT samples.

## A    Additional results on vanilla knowledge distillation

Table 8: Knowledge distillation can improve robustness. The teacher is CLIP-ViT-L/14 @ 224px [48] We use the original knowledge distillation objective [32]. ViT-B [14] and ViT-S [14] are trained with AugReg [59]. Top half of the table is the original performance. Bottom half is the distilled performance. We find that distilling from CLIP can transfer robustness, even on in-distribution data.

| Model | ImageNet | A | C ($\downarrow$) | V2 | Rendition | Sketch | Stylized |
|---|---|---|---|---|---|---|---|
| CLIP [48] | 79.9 | 64.2 | 60.2 | 72.9 | 87.7 | 61.6 | 18.5 |
| ViT-B [59] | 79.9 | 19.0 | 54.5 | 67.9 | 39.5 | 29.2 | 16.6 |
| ViT-S | 77.8 | 11.9 | 63.9 | 66.0 | 36.9 | 25.3 | 12.0 |
| ResNet50 | 76.1 | 0.0 | 76.7 | 63.2 | 36.1 | 24.1 | 7.4 |
| ResNet34 | 66.5 | 3.0 | 94.5 | 54.7 | 32.4 | 21.0 | 5.6 |
| ViT-B | 78.6 | 16.0 | 61.5 | 67.2 | 51.5 | 34.7 | 16.2 |
| ViT-S | 79.3 | 18.1 | 59.1 | 68.8 | 45.9 | 30.6 | 14.3 |
| ResNet50 | 77.8 | 7.4 | 69.0 | 67.6 | 47.0 | 32.3 | 8.7 |
| ResNet34 | 74.5 | 3.6 | 77.1 | 63.2 | 41.2 | 28.5 | 9.3 |
| Average Change | +2.1 | +2.8 | -5.7 | +3.75 | +10.2 | +6.63 | +1.73 |

Table 9: Results on other student/teacher architectures with the original KD objective, on ImageNet-1K. Robustness can be distilled from a variety of robust teachers.

| Model | Teacher | IM | A | C ($\downarrow$) | V2 | Rendition | Sketch | Stylized |
|---|---|---|---|---|---|---|---|---|
| RN50 | ViT-B + DAT | 80.0 | 8.1 | 66.3 | 68.5 | 40.9 | 29.4 | 8.6 |
| RN50 | DrViT-S | 79.3 | 8.2 | 67.4 | 68.4 | 41.7 | 30.0 | 8.9 |
| RN34 | RN50 + AugMix | 72.6 | 1.9 | 80.2 | 61.5 | 37.9 | 25.6 | 8.6 |
| RN34 | DrViT-S | 74.2 | 2.5 | 77.4 | 62.1 | 38.2 | 25.4 | 8.7 |
| RN34 | ViT-B + DAT | 74.3 | 2.5 | 77.1 | 62.3 | 37.8 | 25.4 | 8.4 |
| RN34 | CLIP-RN101 | 72.4 | 3.9 | 79.8 | 61.2 | 45.4 | 29.8 | 8.5 |

In Tab. 8 we find that surprisingly, distilling from CLIP on only in-distribution data is able to transfer robust representations, but is generally outperformed by DAD. This works especially well on smaller models, like ResNet34. In fact, it can also improve clean accuracy compared to training from scratch, for all the models we test except ViT-B. In Tab. 9, we find that distilling from CLIP generally results in the highest average robust performance, especially for natural distribution shifts. However, any robust teacher can distill robustness in this setting, including a ResNet50 trained with AugMix as the only robustness intervention. This matches our results for Tab. 5.

## B    Additional ablations

### B.1    Choice of generative model

We use VQGAN [15] as our generative model of choice due to its nature as a image-to-image model, making it suitable as a discretizer. To justify this, we also experiment with Stable Diffusion [53], a

Table 10: We ablate the use of VQGAN by using Stable Diffusion with DAD. The results are significantly worse, indicating the need to use a image-to-image model to discretize adversarial examples.

|  | ImageNet | V2 | Rendition | Sketch | A | Avg |
|---|---|---|---|---|---|---|
| Stable-Diffusion | 79.1 | 67.8 | 45.9 | 33.4 | 22.0 | 49.6 |
| VQGAN | **79.6** | **69.9** | **65.1** | **46.1** | **31.8** | **69.5** |

Table 11: VQGAN sampling baseline comparison. To show the importance of using model gradients to discover a diverse data augmentation, we sample from the VQGAN without optimizing for a perturbation. The results are significantly worse than DAD.

|  | ImageNet | V2 | R | Sketch | A | Avg |
|---|---|---|---|---|---|---|
| VQGAN - Sample | **80.9** | **70.1** | 49.3 | 34.9 | 24.0 | 51.8 |
| VQGAN - Grad | 79.6 | 69.9 | **65.1** | **46.1** | **31.8** | **69.5** |

text-to-image generative model. We use the generic prompt "A photo of an object". We observe in Tab. 10 a significant decrease in performance when trained using DAD compared to VQGAN. This suggests the importance of using a discretizer for DAD. Perhaps modifying the text prompt for could boost performance and be an interesting avenue for future work, especially since CLIP also requires a text prompt.

## B.2 Use of gradients

DAD is based on adversarial training and uses gradients to find the most diverse and useful data augmentations. To show the importance of using teacher gradients to generate adversarial examples, we implement a sampling-based baseline where we discretize images without the added perturbation. The results in Tab. 11 are significantly worse than DAD, indicating the need to use gradients to discover diverse samples. This is also supported by our theoretical analysis that indicates more diverse adversarial examples are better for robustness. Higher in-distribution performance also suggests the samples are less diverse.

## B.3 Transfer to adversarial robustness

Although we center our study on out-of-distribution robustness, it is natural to examine the effect of DAD on adversarial robustness due to the use of adversarial training. In Tab. 12 we attack our trained models with adversarial attacks of various difficulty. We observe a small improvement in adversarial robustness for simpler attacks, but neither DAT or DAD is robust to AutoAttack. This is because the perturbation is discretized and no longer represents the original adversarial example. We observe that DAT is stronger than DAD for ViT-B. Unlike out-of-distribution robustness, since adversarial robustness is based on perturbations generated with gradients from the base model, DAT models are trained on images closer to these perturbations than DAD models (which were trained on perturbations generated with CLIP gradients). However, for ResNet50, DAD is better even for adversarial robustness as distillation is able to help smaller capacity models learn discrete adversarial examples. We also observe higher ResNet50 performance in general in Tab. 1.

## C   Implementation details

We adopt official hyperparameter settings for a fair comparison for our baselines. For knowledge distillation, we use a temperature of $t = 4$ for all models and $a = 0.5$, following [63]. For DAD, we also weight the second KL-divergence term by $a$. All ViT models are trained with the AugReg [59] hyperparameter and data augmentation configurations.

Table 12: Comparison of adversarial attack methods. We use pretrained models and attack with FGSM [22], PGD [39], and AutoAttack [9]. Models trained on discretized adversarial examples are somewhat robust but fail on stronger attacks.

| Method | FGSM | PGD | AutoAttack |
|---|---|---|---|
| ResNet50 | 23.5 | 1.0 | 0.0 |
| ResNet50 DAT | 33.0 | 5.9 | 0.0 |
| ResNet50 DAD | **43.5** | **12.6** | 0.0 |
| ViT-B | 49.4 | 24.7 | 0.0 |
| ViT-B - DAT | **64.9** | **26.2** | 0.0 |
| ViT-B - DAD | 47.2 | 25.0 | 0.0 |

Following [41], we use the pretrained VQGAN weights from the official GitHub [2]. The VQGAN with f = 8, d = 4 and K = 16384 is used for main experiments.

We use one iteration for the adversarial attack, and an attack learning rate of 0.1.

For the DAT + DAD variant, we add an additional cross entropy loss term with the student adversarial example to the overall training objective, and weight by $a$.

We conduct all of our experiments on 8 32GB NVIDIA V100 GPUs.

## D  Wasserstein distance comparisons

To justify our theoretical framework and empirical results we investigate the relationship between Wasserstein distance and model performance. We use a pretrained ResNet50 and calculate Wasserstein distance from batch norm statistics on different distributions using 1000 mini-batches. These distributions are data augmentations generated with the respective methods. In 2a we find that DAD tends to have better performance the larger the distribution shift. This is likely due to the help of distillation letting the model learn robust representations. Additionally, in 2b, we find that relative to the Wasserstein distance between clean ImageNet images and a distribution shift and baseline models, our method has higher performance.

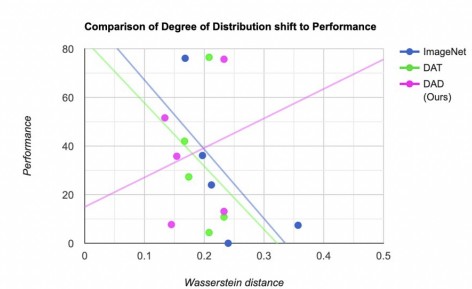

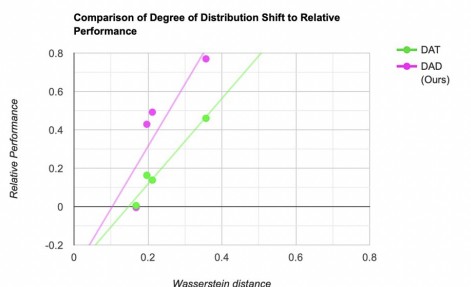

(a) The relationship between Wasserstein distance and performance. DAD is the only method that improves performance when the generated data augmentation is more diverse.

(b) The relative relationship between Wasserstein distance and performance to the baseline model. DAD has stronger performance compared to DAT over the standard model.

## E  Proofs

**Proof of Lemma 3.1.**

[2]https://github.com/CompVis/taming-transformers

*Proof.* Variational divergence $tv$ measures the divergence between distributions, where $\mathcal{B}$ is the set of measurable subsets in $P$ and $P'$

$$tv(P, P') = 2 \sup_{B \in \mathcal{B}} \left| \Pr_P[B] - \Pr_{P'}[B] \right|$$

$$\begin{aligned} r(P') &= r(P') + r(P) - r(P) \\ &\le r(P) + |r(P') - r(P)| \\ &\le r(P) + \int |\delta(x) - \delta'(x)|\, dx \\ &\le r(P) + tv(P, P'). \end{aligned}$$

Following [5] and from Assumption 2, we can bound total variation with Wasserstein's distance. We take $K : \mathbb{R} \to \mathbb{R}$, a kernel satisfying a suitable moment condition, so for any coupling of $P$ and $P'$,

$$\| K * p - K * p' \|_\epsilon \le \sup_{s \ne t} \frac{\|T_s(K) - T_t(K)\|_r}{|s - t|} W_\epsilon(P, P').$$

by the Jensen's inequality. Assume that $p, p' \in H_1^\alpha(\mathbb{R})$. Let $\{\phi_m\}$ be the orthonormal system in $L^2([-1, 1])$ of Legendre polynomials defined by

$$\phi_0(x) = 2^{-1/2} I(|x| \le 1), \quad \phi_m(x) = \sqrt{\frac{2m+1}{2}} \frac{1}{2^m m!} \frac{d^m}{dx^m}[(x^2 - 1)^m] I(|x| \le 1),$$

for $x \in \mathbb{R}$ and $m \in \mathbb{N}$. Define

$$K(x) = \sum_{m=0}^{\alpha} \phi_m(0)\phi_m(x).$$

Then, by Propositions 4.1.5 and 4.1.6 of [19],

$$\|Kh * p - p\|_1 \preccurlyeq \|p\|^{H_1^\alpha} h^\alpha \quad \text{and} \quad \|Kh * p' - p'\|_1 \preccurlyeq \|p'\|^{H_1^\alpha} h^\alpha.$$

Since $\max_{x \in [-1,1]}(|\phi_m(x)| \vee |\phi'_m(x)|)$ is bounded by a constant depending only on $m$, where $\phi'_m$ is the derivative of $\phi_m$ and $a \vee b$ is the maximum of $a$ and $b$,

$$\begin{aligned} \|T_s(\phi_m) - T_t(\phi_m)\|_1 &= \int |\phi_m(x - s) - \phi_m(x - t)| dx \\ &\le \int_{\{|x-s| \vee |x-t| \le 1\}} |\phi_m(x - s) - \phi_m(x - t)| dx + 2|s - t| \max_{x \in [-1,1]} |\phi_m(x)| \\ &\le 4|s - t| \max_{x \in [-1,1]} |\phi'_m(x)| + 2|s - t| \max_{x \in [-1,1]} |\phi_m(x)| \\ &\preccurlyeq |s - t|. \end{aligned}$$

Thus,

$$\|T_s(Kh) - T_t(Kh)\|_1 \le \frac{1}{h} \sum_{m=0}^{\alpha} \phi_m(0) \int \left| \phi_m\left(\frac{x-s}{h}\right) - \phi_m\left(\frac{x-t}{h}\right) \right| dx \preccurlyeq \frac{|s-t|}{h},$$

where $\phi_m(0)$ and $\alpha$ depends only on $\alpha$. By the triangle inequality, we have

$$\begin{aligned} \|p - p'\|_1 &\le \|p - Kh * p\|_1 + \|Kh * p - Kh * p'\|_1 + \|Kh * p' - p'\|_1 \\ &\preccurlyeq \|p\|^{H_1^\alpha} h^\alpha + \frac{W_1(P, P')}{h} + \|p'\|^{H_1^\alpha} h^\alpha. \end{aligned}$$

If we take

$$h = \left( \frac{W_1(P, P')}{\|p\|^{H_1^\alpha} + \|p'\|^{H_1^\alpha}} \right)^{1/(\alpha+1)},$$

the proof is complete. $\qquad \square$

**Proof of Lemma 3.3.**

*Proof.* Let $(Z, d_Z)$ be a metric space where $Z = X \times Y$ and $d_Z$ is defined as:
$$d_Z(z, z_0) = d_Z((x, y), (x_0, y_0)) = (d_X(x, x_0) + d_Y(y, y_0))$$
where $d_X$ and $d_Y$ represent the metric in the feature space and label space, respectively. Then we can define the Wasserstein distance between $P$ and $P^*$,
$$W_p(P, P^*) := \inf_{M \in \Gamma(P, P^*)} \mathbb{E}_{(z, z_0) \sim M}[d_Z(z, z_0)],$$
where $\Gamma(P, P^*)$ denotes the collection of all measures on $Z \times Z$ with marginals $P$ and $P^*$ on the first and second factors, respectively.

Following [65], we use the minimax approach, considering the worse-case $P^*$ in the Wasserstein ball $\mathcal{B}_\varepsilon^{w_p}$ of radius $\varepsilon$ centered around $P$ where
$$\mathcal{B}_\varepsilon^{w_p}(P) = \{P^* : w_p(P, P^*) \leq \varepsilon\}$$

Next we define a transport map $T : Z \to Z$ to push $P$ to $P^*$ as follows:
$$z = (x, y) \mapsto (x^*, y)$$
where $x^* = \arg\max_{x_0 \in P^*} l(\theta(x_0), y)$. By the definition of $d_Z$, $d_Z((x, y), (x^*, y)) = d_X(x, x^*)$.

Finally, let $P^* = T_\theta \# P$, the pushforward of $P$ by $T_\theta$, then we have $R(P, \epsilon) = R(P^*)$. By the definition, we have
$$\begin{aligned} R(P, \epsilon) &= \mathbb{E}_{(x,y) \sim P}[\max\ l(\theta(x_0), y)] \\ &= \mathbb{E}_{(x,y) \sim P}[l(\theta(x^*), y)] \\ &= \mathbb{E}_{(x,y) \sim P^*}[l(\theta(x), y)]. \end{aligned}$$

Therefore, $r(P, \epsilon) = r(P^*)$. This lets us establish upper bound on the worst-case in the Wasserstein ball and bound the adversarial expected risk. Next we define the radius of the adversary constrained by $B$ as $\varepsilon_B := \sup_{x \in B} d_X(x, 0)$. For any hypothesis $h$ and the corresponding $P^* = T_\theta \# P$, we have $w(P, P^*) \leq \varepsilon_B$. By the definition of the Wasserstein distance, we have
$$\begin{aligned} w(P, P^*) &\leq \mathbb{E}_P[d_Z(Z, T_\theta(Z))] \\ &= \mathbb{E}_P[d_X(x, x^*)] \\ &\leq (\varepsilon_B), \end{aligned}$$
where the last inequality uses the translation invariant property of $d_X$. Therefore, we have
$$w(P, P^*) \leq \varepsilon_B.$$

$\square$

**Proof of Lemma 3.4**

*Proof.* We use $P$ to denote the (at least) one distribution in the intersection of $\mathbf{P}_1$ and $\mathbf{P}_2$.
$$\mathbb{E}_{\hat{P}}\left[w(D(\phi(\mathbf{P}_1)), P') - w(D(\phi(P)), P')\right]$$
$$\begin{aligned} &\leq \mathbb{E}_{\hat{P}}\left[w(\mathbf{P}_1, P_1) + w(\mathbf{P}_1, P')\right] - \mathbb{E}_{\hat{P}}\left[w(P, P_2) - w(P, P')\right] \\ &= \mathbb{E}_{\hat{P}}\left[w(\mathbf{P}_1, P_1) - w(P, P_2)\right] + \mathbb{E}_{\hat{P}}w(\mathbf{P}_1, P') + w(P, P') \\ &= \mathbb{E}_{\hat{P}}\left[\sup_{P \in \mathbf{P}} r_{\text{worst}}(P, \epsilon) - r_{\text{worst}}(P, \epsilon)\right] + \mathbb{E}_{\hat{P}}w(\mathbf{P}, P') + w(P, P') \\ &\leq \sup_{P \in \mathbf{P}_1} r_{\text{worst}}(P, \epsilon) - r_{\text{worst}}(P, \epsilon) + \mathbb{E}_{\hat{P}} \sup_{P \in \mathbf{P}} w(P, P') + w(P, P') \\ &= 2\mathbb{E}_{\hat{P}} \sup_{P \in \mathbf{P}_1} w(P, P') + c, \end{aligned}$$
where $c$ is a positive constant.

Similarly, we can have
$$\mathbb{E}_{\hat{P}}\left[w(D(\phi(\mathbf{P}_2)), P') - w(D(\phi(P)), P')\right] \geq c - 2\mathbb{E}_{\hat{P}} \sup_{P \in \mathbf{P}_2} w(P, P').$$

$\square$

# F    Visualizations

(a) Clean       (b) Pixel AT       (c) DAT       (d) DAD

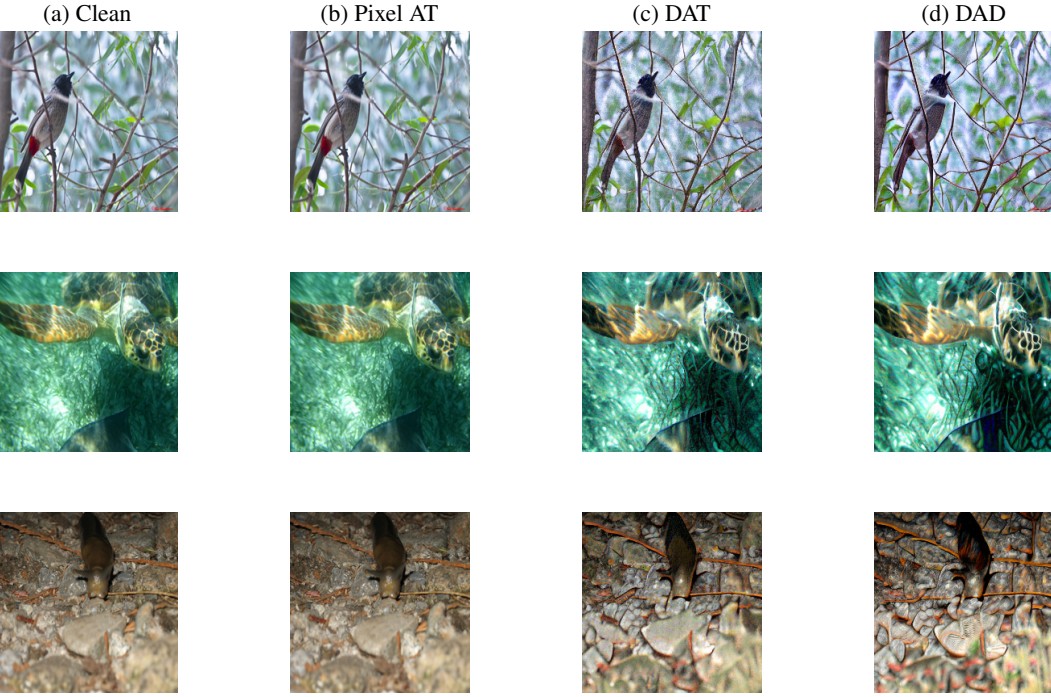

Figure 3: Additional visualizations of generated images. To highlight the difference, we use adversarial examples that are classified differently by the base model. Using CLIP in DAD results in a more diverse adversarial example than a vanilla ResNet50. Adversarial examples in pixel-space are imperceptible.

