# OpenReview forum: "Distilling Out-of-Distribution Robustness from Vision-Language Foundation Models"
_NeurIPS.cc/2023/Conference — NeurIPS 2023 poster_

### Official Review · Reviewer_ktZ5 · 2023-07-05

**Soundness:** 2 fair
**Presentation:** 1 poor
**Contribution:** 2 fair
**Rating:** 5
**Confidence:** 3

**Summary:**

The authors propose a framework that improves the image classification model's robustness by distilling CLIP models and augmenting adversarial learning with pre-trained generative models. The authors follow classical adversarial learning to generate perturbed examples and then input the examples to VQ-GAN. Lastly, they distill a CLIP model into a smaller network with both normal and augmented adversarial examples. While I have some concerns about the results, extensive experiments verify the effectiveness of the proposed method in many settings.

**Strengths:**

1. The extensive experiments cover various settings, such as different architectures for teacher and student models and variations of ImageNet.
2. The authors provide some theory explaining that a more diverse distribution could be helpful for distillation.

**Weaknesses:**

1. The research problem is unclear. In L37, the authors claim to "train small robust models with a large-scale model as a teacher, but without access to the teacher's training data." However, they focus more on out-of-distribution robustness in the later parts and the experiments.

2. The proposed method needs to be better justified. In L118, the authors suddenly introduce generative models to augment adversarial examples and claim that it helps generalizability without any reasoning. If possible, the authors should provide more motivation and insights into it and conduct some experiments.

3. Many notations are not defined. For example, the definition of $\theta(x, x\prime)$ is missing. What exactly does $Q(x\prime)$ work?

4. The novelty is limited. The main contribution seems solely to leverage VQ-GAN to enhance distillation. The reason for choosing VQ-GAN is also missing.

5. The research scope is limited. The authors claim in L45 that distilling large-scale models into smaller models improves adversarial robustness. However, the experiments only consider CLIP models.

**Questions:**

1. The performance could be better, even though it outperforms prior work. In Table 1, neither the baselines nor the proposed method achieves the CLIP baseline performance. Maybe one should look into cross-modality instead of augmenting query datasets for distillation.

2. What does it mean by *DAD (no distill)*? I recommend the authors refer the readers to the corresponding loss terms.

3. The proposed method is similar to leveraging more data to perform distillation (Eq. 8). The author may like to explore using different kinds of data (e.g., ImageNet + CIFAR-10 or ImageNet + ImageNet with augmentation), which could even outline the contribution of the proposed method.

**Limitations:**

1. The authors only consider distilling CLIP models.

---

> ### Author Rebuttal · Authors · 2023-08-10
>
> We thank the reviewer for the insightful comments. We will address the concerns about the novelty and justification for our method.
> \
> \
> *The research problem is unclear...they focus more on out-of-distribution robustness in the later parts and the experiments.*
>
> Thank you for pointing this out. To clarify, the goal of our DAD is to improve out-of-distribution robustness on smaller models and datasets by distilling from a large-scale teacher. We point out the distinction between different forms of robustness on L67-69, but agree with the reviewer that this should be clear from the beginning and will revise the statement in the updated paper.
> \
> \
> *The proposed method needs to be better justified...If possible, the authors should provide more motivation and insights into it and conduct some experiments.*
>
> The choice of data augmentation is one of the key components of our proposed method. Since we focus on out-of-distribution robustness, we adopt a generative model to generate data augmentations adversarially to ensure the resulting transformation is semantically meaningful. The VQGAN can apply a semantically meaningful transformation on the image to better capture a useful distribution shift than standard pixel-level perturbations. The effects of not applying a generative model and using common data augmentations can be observed in Table 7 in the Appendix. Here we only used Mixup and RandAugment and demonstrate the results of distilling from CLIP without using the generative model / adversarial examples. The results are much worse, with an average -13.2% decrease in performance on natural distribution shifts.
> \
> \
> *Many notations are not defined. For example, the definition of theta(x,x’) is missing. What exactly does Q(x’) work?*
>
> X refers to the base input (L101) and x’ refers to the augmented input (L104). Theta(x, x’) means passing the image or augmented image through the student model. We will add this notation to the revised paper. Q(x’) indicates the process of discretizing an input image, which we describe in more detail in L145-152. To summarize, after we apply a normal perturbation x -> x’, the VQGAN encodes input x’ into a latent representation which is then decoded, applying a semantic transformation.
> \
> \
> *The novelty is limited. The main contribution seems solely to leverage VQ-GAN to enhance distillation. The reason for choosing VQ-GAN is also missing.*
>
> We kindly emphasize our key contributions below:
> 1. *Setting.* We are the first to also introduce a robust teacher and use knowledge distillation to further improve robustness. We focus on OOD-robustness due to practical usefulness and the specialization of large-scale models on natural distribution shifts. These models exhibit strong robustness due to their diverse training data, and we are the first to leverage these representations as an additional form of regularization
> 2. *Objective.* We introduce a novel knowledge distillation objective for the OOD-robustness setting where we add a second KL-divergence term between the student and teacher predictions on the augmented image. Previous works in defensive distillation like ARD use the teacher predictions on the normal image for this second term, but this is not adaptive to semantic transformations.
> 3. *Data augmentation.* We indeed use a VQGAN for the data augmentation. However, our key novelty here is using the teacher’s gradients to generate adversarial examples. We find that this results in more diverse adversarial examples closer to the teacher’s representations, and our theoretical framework is based on this idea. We choose VQGAN in particular following the prior SOTA DAT, which finds this to lead to the best result. We also try using the newer Stable Diffusion model in the response to Reviewer qyW3, and find that it results in worse performance.
>
> \
> *The research scope is limited. The authors claim in L45 that distilling large-scale models into smaller models improves adversarial robustness. However, the experiments only consider CLIP models.*
>
> We focus our analysis on CLIP because of its accessibility and performance. At the time of submission, there aren’t any open-source large-scale vision models better than CLIP. However, we note that DAD is agnostic to the choice of teacher. In fact, we provide results in Table 4 using different teacher models. Generally we find that it is better to use CLIP. We hope to use newer foundation models as they become available.
> \
> \
> *The performance could be better, even though it outperforms prior work...Maybe one should look into cross-modality instead of augmenting query datasets for distillation.*
>
> We propose to use CLIP as an additional source of regularization in lower compute regimes, rather than improving upon CLIP. Data augmentation is easy to apply and can be done across a variety of settings, including in cross-modal training. DAD also improves upon prior data augmentations, especially on natural distribution shifts.
> \
> \
> *What does it mean by DAD (no distill)?...*
>
> This means training with only the L1 cross-entropy loss but with DAD samples, which corresponds to standard training using previously computed DAD samples for augmentation. We will make this clearer in the revised paper.
> \
> \
> *The proposed method is similar to leveraging more data to perform distillation (Eq. 8). The author may like to explore using different kinds of data...*
>
> Indeed we do try simpler data augmentations in Table 7 in the Appendix. Here we use Mixup and RandAugment and find the results are worse. The goal of using the VQGAN and teacher-gradient-based adversarial examples is to generate semantically meaningful and diverse perturbations. We find that these simpler forms of data do not represent difficult real-world transformations as well as DAD. Combining different datasets is certainly an interesting direction to consider.
> \
> \
> We hope our response addressed the reviewer’s questions and concerns. We are happy to answer any further questions.

---

> > ### Comment · Reviewer_ktZ5 · 2023-08-11
> > **Response to the Authors**
> >
> > I would like to thank the authors for the detailed response.
> >
> > After reading the response and carefully re-reading the paper, my concern about the research problem in this paper is partially addressed. However, the presentation of the paper remains a main issue. Many descriptions in this paper are unclear or even confusing. I would like to clarify my understanding below before adjusting my score.
> >
> > 1. In L21-26, it looks like the authors plan to work on improving the in-distribution accuracy under adversarial learning. But, it turns out to investigate data augmentation with the *zero-shot generalization* of foundation models (L33).
> > 2. L36-38, again, is confusing. As mentioned in my previous comment, it gives the impression that the goal is solely to distill large models into smaller ones.
> > 3. Is L40-L46 a finding from the authors or cited from other works? Does this only happen to CLIP?
> > 4. Following (3), Table 7 in the appendix is important evidence to motivate the proposed method, where normal augmented data are unsuitable for distilling CLIP models. It is unclear why the authors leave them in the appendix.
> > 5. The work's main contribution is to leverage discretizers, i.e., VQ-GAN, and the idea is highly built upon the prior work DAT. However, relations, detailed comparisons, or motivations are totally missing in this work, making the readers hard to judge the contribution. The operation of VQ-VAE is also missing.
> > 6. Math presentations need to be revised. For instance, where does U come from in L101? Q sometimes takes one input argument but sometimes 2.
> >
> > I might change the score in the end, but I still find the writing not ready for publication, despite its effectiveness. I strongly recommend rephrasing the introduction if the paper gets accepted.

---

> > > ### Author Response · Authors · 2023-08-13
> > > **Response to the Reviewer**
> > >
> > > We thank the reviewer for the response. We are glad that our rebuttal has addressed the reviewer's concerns especially on the research problem. It is also encouraging to hear that the reviewer is open to adjusting the score. The remaining concerns from the reviewer are on the presentation, and we thank the reviewer for clarifying them. Below we address all these points in further detail, and we will revise the paper accordingly.
> > >
> > > \
> > > *In L21-26, it looks like the authors plan to work on improving the in-distribution accuracy under adversarial learning. But, it turns out to investigate data augmentation with the zero-shot generalization of foundation models (L33).*
> > >
> > > In this section, our intent was to establish the motivation for DAD. We kindly refer the reviewer to L36, where our motivation is explained as combining the strengths of regularization and foundation models. Before L36, we point out the weaknesses of either approach. In L21-26, we argue that prior regularization techniques like adversarial training reduces in-distribution accuracy (L25) or does not transfer well to out-of-distribution robustness (L26). Therefore, our focus is on distilling foundation models through data augmentation. We realize that the current presentation in paragraph L21-26 could be confusing. We will revise this paragraph to make it clearer by 1) highlighting the deficiencies of current data augmentation methods in particular, and 2) moving the discussion on adversarial robustness to the Related Work section.
> > >
> > > \
> > > *L36-38, again, is confusing. As mentioned in my previous comment, it gives the impression that the goal is solely to distill large models into smaller ones.*
> > >
> > > We see the reviewer’s concern. For clarity, it may be helpful to separate the methodology from the goal. Broadly, our goal is indeed to improve out-of-distribution robustness, as the reviewer previously noted. But our novelty and methodology are based on distilling from large models. We will make this clearer by revising L36-38 in the following way:
> > >
> > > - In this paper, we aim to connect these two lines of work. Our goal is to improve the out-of-distribution robustness of small models without large-scale training. To this end, we introduce a foundation model as a teacher to improve the diversity of data augmentation and directly distill robust representations.
> > >
> > > \
> > > *Is L40-L46 a finding from the authors or cited from other works? Does this only happen to CLIP?*
> > >
> > > The finding is from us. The ability to transfer robustness using vanilla knowledge distillation is our finding and unexplored in prior works.
> > >
> > > The finding does not only happen to CLIP. We observe in Table 8 in the Appendix that this finding generalizes across different teacher architectures.
> > >
> > > \
> > > *Following (3), Table 7 in the appendix is important evidence to motivate the proposed method, where normal augmented data are unsuitable for distilling CLIP models. It is unclear why the authors leave them in the appendix.*
> > >
> > > We left this result in the Appendix due to lack of space, but we agree that it is important. So we will add one of the main results of Table 7 (CLIP can distill robustness) as a figure in the introduction, and we will also place it in the experimental section as an ablation for using common data augmentations to demonstrate the need for the generative model.

---

> > > ### Author Response · Authors · 2023-08-13
> > > **Response to the Reviewer cont.**
> > >
> > > **continued from the previous response*
> > >
> > > \
> > > *The work's main contribution is to leverage discretizers, i.e., VQ-GAN, and the idea is highly built upon the prior work DAT.*
> > >
> > > We would like to respectfully clarify that although we do draw upon the idea of using VQ-GAN from DAT, we additionally contribute a new setting, knowledge distillation objective for out-of-distribution robustness, and data augmentation generated using teacher gradients. We refer the reviewer to the initial response to concerns about novelty for more detail.
> > >
> > > *However, relations, detailed comparisons, or motivations are totally missing in this work, making the readers hard to judge the contribution.*
> > >
> > > - We already provided relations to prior uses of generative models for regularization in L79-83.
> > > - The motivation for using a discretizer in our setting can be found in L117-121.
> > > - While we do not provide detailed comparisons to other discretizers in the paper, we refer in L145-146 that this is due to the direct use of VQ-GAN from DAT where this is already ablated in Table 5 [1]. In the rebuttal phase, we also previously provided an additional result for Reviewer qyW3 on Stable Diffusion that demonstrates the need for VQ-GAN, and we will add this result as an additional ablation in the revision. We also list the result here:
> > >
> > > | | ImageNet | V2 | R | Sketch | A | Avg |
> > > | --- | --- | --- | --- | --- | --- | --- |
> > > | Stable Diffusion | 79.1|	67.8	|45.9	|33.4	| 22.0| 49.6 |
> > > | VQGAN  | 79.6 | 69.9 | 65.1 | 46.1 | 31.8 | 69.5 |
> > >
> > > \
> > > *The operation of VQ-VAE is also missing.*
> > >
> > > We described the operation of VQ-GAN in L145-150. For additional clarity, we will add details in the revised paper about how VQ-GAN is trained, why an image-to-image discretizer is necessary in particular, and how it is used in DAD to generate adversarial examples.
> > >
> > > \
> > > *Math presentations need to be revised. For instance, where does U come from in L101? Q sometimes takes one input argument but sometimes 2.*
> > >
> > > We define U as the set of all transformed images (L101-102). We checked the relevant sections and did not find any particular instances where Q takes more than one input, does the reviewer mean $\theta$ ? We use two input arguments for simplicity for the student model $\theta$ in Eqs. 4 and 5. We realize this could be confusing and will expand the equation to include terms for both the clean and perturbed inputs like so.
> > >
> > > Eq. 4
> > >
> > > min $E_{(x,y)\sim P}~[ ~l(\theta(x), y) ~+ $ max $l(\theta(x'), y) ~]$
> > >
> > > Eq. 5
> > >
> > > min $E_{(x,y)\sim P}~[ ~l(\theta(x), y) ~+ $ max $l(\theta(Q(x')), y) ~]$
> > >
> > > \
> > > We hope our response addressed the reviewer’s remaining concerns. We are happy to answer further questions.
> > >
> > > \
> > > [1] Xiaofeng Mao, Yuefeng Chen, Ranjie Duan, Yao Zhu, Gege Qi, Shaokai Ye, Xiaodan Li, Rong Zhang, and Hui Xue. Enhance the visual representation via discrete adversarial training. arXiv preprint arXiv:2209.07735, 2022.

---

> > > ### Author Response · Authors · 2023-08-19
> > > **Follow up to the Reviewer**
> > >
> > > Hello,
> > >
> > > As the discussion period is closing soon, we’d like to follow up and see if the Reviewer has had a chance to consider our response. We hope the Reviewer can raise the score if the concerns were addressed.

---

> > > > ### Comment · Reviewer_ktZ5 · 2023-08-20
> > > >
> > > > Dear authors,
> > > >
> > > > Thanks for the detailed follow-up. I have adjusted my score accordingly. And, yes, I meant to $\theta$. After the rebuttal, I still strongly recommend the authors revise the introduction and the paper accordingly, outlining the paper's main focus.

---

> > > > > ### Author Response · Authors · 2023-08-21
> > > > >
> > > > > We thank the reviewer for adjusting the score and for providing detailed feedback. We find the comments constructive and will revise the paper accordingly, along with the feedback from other reviewers.

---

### Official Review · Reviewer_qyW3 · 2023-07-06

**Soundness:** 3 good
**Presentation:** 3 good
**Contribution:** 3 good
**Rating:** 5
**Confidence:** 4

**Summary:**

This paper proposes a simple and lightweight framework for improving the robustness of vision models through knowledge distillation. This paper applies pre-trained teacher model to generate adversarial examples and apply VQGAN as a data augmentation method to generate more informative adversarial samples. This paper provides a theoretical framework for applying robust teacher in the knowledge distillation with data augmentation.

**Strengths:**

+The paper applies a large pre trained model based on knowledge distillation to enhance the model's out-of-distribution robustness with data augmentation, this idea is novel.

+The paper provides a theoretical analysis of how to best distill robustness from a robust teacher trained on a large-scale dataset.

+The experiment of the paper can prove the effectiveness of the method.


**Weaknesses:**

-The motivation for applying VQGAN is not so clear. The article claims that VQGAN can discretize adversarial examples, but can other data augmentation technology such as common data augmentation methods (translation, cropping), other variants of GAN, or Stable Diffusion achieve better performance? This part seems to be a direct application, and I think a discussion of the necessity should be provided.

-Although this paper has a pretty performance of out-of-distribution robustness, I wonder about the performance of adversarial robustness (such as PGD or AutoAttack).

-The frame diagram is somewhat rough and crude (this will not affect the final rating, but I hope the author can make some modifications).


**Questions:**

see the weakness

**Limitations:**

see the weakness

---

> ### Author Rebuttal · Authors · 2023-08-10
>
> We thank the reviewer for the helpful suggestions and positive review. We are glad the reviewer found our idea novel and think the experiments and theory prove its effectiveness.
> \
> \
> *The motivation for applying VQGAN is not so clear. The article claims that VQGAN can discretize adversarial examples, but can other data augmentation technology such as common data augmentation methods (translation, cropping), other variants of GAN, or Stable Diffusion achieve better performance? This part seems to be a direct application, and I think a discussion of the necessity should be provided.*
>
> The motivation for applying VQGAN is to ensure perturbations are semantic to better match real-world transformations. This allows us to focus on an out-of-distribution robustness setting. This is motivated by several works that use generative models to construct data augmentations; we adapt the choice VQGAN from the prior SOTA DAT, which finds VQGAN performs best. The VQGAN can apply a semantically meaningful transformation on the image to better capture a useful distribution shift than standard pixel-level perturbations.
>
> The effects of not applying a generative model and using common data augmentations can be observed in Table 7 in the Appendix. Here we use Mixup and RandAugment which are components of AugReg and demonstrate the results of distilling from CLIP without using the generative model / adversarial examples. The results are much worse, with an average -13.2% decrease in performance on natural distribution shifts.
>
> Following the reviewer's suggestion, we also provide the results on Stable Diffusion, a newer model since DAT was proposed. We use the generic prompt "A photo of an {object}". We observe a significant decrease in performance when trained using DAD compared to VQGAN. We hypothesize that VQGAN is better suited for the image-to-image task of discretizing images than a text-to-image model. Perhaps modifying the text prompt for could boost performance and be an interesting avenue for future work, especially since CLIP also requires a text prompt.
>
> | | ImageNet | V2 | Rendition | Sketch | A | Avg |
> | --- | --- | --- | --- | --- | --- | --- |
> | Stable-Diffusion | 79.1 | 67.8 | 45.9 | 33.4 | 22.0 | 49.6 |
> | VQGAN | 79.6 | 69.9 | 65.1 | 46.1 | 31.8 | 69.5 |
>
> \
> \
> *Although this paper has a pretty performance of out-of-distribution robustness, I wonder about the performance of adversarial robustness (such as PGD or AutoAttack).*
>
> This is an interesting observation. We evaluate our ResNet50 checkpoint on FGSM. We observe an improvement on adversarial robustness compared to both standard training and DAT. We will provide results on DamageNet and ViT-B in the discussion phase.
>
>
> | Method       | FGSM  |
> |--------------|-------|
> | ResNet50     | 23.5% |
> | ResNet50-DAT | 33.0% |
> | ResNet50-DAD | 43.5% |
>
> \
> *The frame diagram is somewhat rough and crude (this will not affect the final rating, but I hope the author can make some modifications).*
>
> Thank you for pointing this out, we have uploaded an updated figure in the uploaded PDF.
> \
> \
> We hope our response addressed the reviewer’s questions and concerns. We are happy to answer any further questions or provide additional results.

---

> ### Author Response · Authors · 2023-08-13
> **Follow up on adversarial robustness ablation**
>
> The reviewer asked about the adversarial robustness of models trained with DAD. We previously offered a simple result where there is improved small improvement in robustness against FGSM for ResNet50 and now have results on PGD and AutoAttack and on ViT-B. Below is the updated table with a more comprehensive evaluation.
>
> | Method | FGSM | PGD | AutoAttack |
> | --- | --- | --- | --- |
> | ResNet50 | 23.5 | 1.0 | 0.0 |
> | ResNet50 DAT | 33.0 | 5.9 | 0.0 |
> | ResNet50 DAD | 43.5 | 12.6 | 0.0 |
> | ViT-B | 49.4 | 24.7 | 0.0 |
> | ViT-B - DAT | 64.9 | 26.2 | 0.0 |
> | ViT-B - DAD | 47.2 | 25.0 | 0.0 |
>
> There is a small improvement in adversarial robustness for simpler attacks, but neither DAT or DAD is robust to AutoAttack as the VQ-GAN discretizes the perturbation. We observe that DAT is stronger than DAD for ViT-B. Unlike out-of-distribution robustness, since adversarial robustness is based on perturbations generated with gradients from the base model, DAT models are trained on images closer to these perturbations than DAD models (which were trained on perturbations generated with CLIP gradients). However, for ResNet50, DAD is better  even for adversarial robustness as distillation is able to help smaller capacity models learn discrete adversarial examples, like we observe in Table 1.
>
> We hope this result is helpful and better answers the reviewer's question. We would be happy to answer any further questions.

---

### Official Review · Reviewer_Y3Hr · 2023-07-06

**Soundness:** 3 good
**Presentation:** 3 good
**Contribution:** 3 good
**Rating:** 6
**Confidence:** 4

**Summary:**

The paper introduces a new method to improve the robustness of vision models through knowledge distillation by leveraging a large-scale pre-trained teacher model (CLIP) and a VQ-GAN discretizer. The paper does a great job motivating the method and arguing for knowledge distillation when tackling out of distribution robustness. It also includes extensive theoretical justifications. And finally, the paper introduces and evaluates a novel distillation objective ("DAD"). In the proposed method ("DAD") the student model is trained to minimize a loss made up of a standard cross-entropy loss, a KL to the teacher's predictions (i.e. standard knowledge distillation) and an additional KL between the teacher and student's predictions on additional (teacher-adversarial) data points (which is novel). The paper also includes a reasonable evaluation section for the method with overall positive results, while highlighting some considerable performance improvements on natural distribution shifts over baselines.

**Strengths:**

The paper is relatively well written and easy to follow. The paper introduces a relatively novel and interesting knowledge distillation loss which should have increasingly wider applicability and grow in significance as more and more of the ML world is moving towards leveraging pre-trained foundation models.

**Weaknesses:**

- Missing paragraph detailing DAT: since a large proportion of the strongest results seem to be relying on combining DAD with DAT. The paper should also include a description of DAT's loss and the "DAT + DAD" loss written out (in the appendix if no space available in the main body).
- The paper mentions the method has "negligible computational overhead" -- I strongly disagree. Computing adversarial examples for a CLIP model that also passes them through VQGAN is not negligible. One can say that the computation cost is amortized though -- as you only have to do it once and then read it from disk.
- Missing "Avg" column in Table 2 and 3, and best results are not shown in bold in Table 2.
- Loss is unclear (Eq 9) -- please expand and clarify this to show which variables are being optimized over, at which stage, which are the constraints exactly, and specify the losses (what is l_1 and l_2 precisely - these are only mentioned in passing in line 125-126 being "typically"/"usually" CE and KL); and also please use or remove "n", as currently it is not used to index anything.

- Missing references, e.g., on [1] (on worst case risk over a population of distributions) and [2] (for a related formulation of a similar set of assumptions and similar theoretical conclusion):

[1] Arjovsky et al., Invariant Risk Minimization.

[2] Calian et al., Defending against image corruptions through adversarial augmentations.

- Figure 1 does not seem to be faithful depiction of the method as described in the paper, so it should ideally be improved significantly (the caption could also give more details towards explaining the method).

**Questions:**

- At a high level, looking at equation (9) of the full model loss, I'm trying to understand how each component affects downstream robustness. So, I'd like to able to map the contribution of each of the two "l_2" losses and the way in which the adversarial example is computed to specific results. This seems like a pretty straightforward and important ablation. Table 3 and Table 5 present just a slice of this ablation as far as I can tell. Could you please clarify how one can reconstitute this ablation from the results presented in the paper?

- The adversarial examples could also not be computed but rather perturbations could be sampled instead (while still using the VQGAN), or random data augmentation methods (or just an identity function used, i.e. which would correspond to just dropping the last l_2 loss in Eq 9). (From Table 6 it seems that more iterations actually results in worse performance across many metrics.) Have you experimented with this?

- In Table 1, does the row "DAT + DAD (Ours)" refer to training with DAT & DAD on top of AugReg-ViT[55]? If yes, please mark it with a leading "+" for clarity.

- In line 303, the paper states "These adversarial examples are generated with DAD" -- was "DAT" intended?

**Limitations:**

The authors adequately addressed the method's limitations.

---

> ### Author Rebuttal · Authors · 2023-08-10
>
> We thank the reviewer for the extensive and positive review and helpful suggestions. We are glad the reviewer found our proposed method novel and relevant.
> \
> \
> *Missing paragraph detailing DAT: since a large proportion of the strongest results seem to be relying on combining DAD with DAT...*
>
> Thank you for bringing this to our attention, we will formally write out the objectives for all compared methods in the revised paper. DAT follows the standard adversarial training objective, but with discretized samples:
> \
> \
> $CE(x, y) + CE(x’,y)$
> \
> \
> We combine DAT with DAD by adding the second L1 objective from DAT to the DAD objective. The full loss is:
> \
> \
> $CE(x, y) + KL(\theta(x), \phi(x)) + KL(\theta(x’), \phi(x’)) + CE(x’,y)$
> \
> \
> \
> *The paper mentions the method has "negligible computational overhead" -- I strongly disagree. Computing adversarial examples for a CLIP model that also passes them through VQGAN is not negligible. One can say that the computation cost is amortized though -- as you only have to do it once and then read it from disk.*
>
> We apologize for not making the claim clearer. We agree that generating adversarial examples with a VQGAN is not negligible. But as the reviewer suggests, the cost over a full training run is amortized. We generate the set of adversarial examples once from the frozen teacher and reuse it for all of our training. For training a ViT-B model for 300 epochs, this cost is essentially reduced by a factor of 300. We also use the straight-through gradient applied in DAT, so the actual cost of generation is the same, but because DAT generates these examples every epoch, DAD is significantly cheaper. For a more formal analysis, we follow the methodology in FastAdvProp [1]. The only major additional cost is training on the new adversarial examples. We will also update the claim in the revised paper to make the statement more explicit.
>
> | Method               | Attack Steps | Training Budget |
> |----------------------|--------------|-----------------|
> | ImageNet | 0           | 1x             |
> | Adversarial Training | 10           | 11x            |
> | AdvProp              | 5            | 7x              |
> | AdvProp              | 1            | 3x              |
> | DAT                  | 1            | 3.5x            |
> | DAD                  | 1            | 2x              |
>
> \
> *Missing "Avg" column in Table 2 and 3, and best results are not shown in bold in Table 2.*
>
> We have uploaded an updated table in the PDF and will include this in the revised paper.
> \
> \
> *Loss is unclear (Eq 9)...and also please use or remove "n", as currently it is not used to index anything.*
>
> We apologize for the confusion. We will remove “n” from this loss. We provide the loss here.
>
> $CE(x, y) + KL(\theta(x), \phi(x)) + KL(\theta(x’), \phi(x’))$
> \
> \
> *Missing references...*
>
> Thank you for bringing these references to our attention. We will cite them in the revised paper.
> \
> \
> *At a high level, looking at equation (9) of the full model loss, I'm trying to understand how each component affects downstream robustness...Could you please clarify how one can reconstitute this ablation from the results presented in the paper?*
>
> We apologize for not making the objectives of the baselines and ablations explicit. We will add this to the revised paper. Yes, the results from ablating different components of DAD can be reconstruction from our experiments.
> \
> \
> $L1$
>
> $CE(x, y)$
>
> This corresponds to standard training on ImageNet and is the simplest baseline we compare to in Table 1.
> \
> \
> $L1 + L2_1$
>
> $CE(x, y) + KL(\theta(x), \phi(x))$
>
> This corresponds to standard knowledge distillation. We observe the results of this objective on various student/teacher architectures in Table 7. We find that this simple objective is enough to transfer a degree of robustness from a variety of teachers.
> \
> \
> $L1 + L2_1 + L2_2$
>
> $CE(x, y) + KL(\theta(x), \phi(x)) + KL(\theta(x’), \phi(x’))$
>
> This is our proposed method DAD.
> \
> \
> $CE(x, y) + CE(x’, y)$
>
> To separate L2_2 and see the direct effect of using DAD adversarial examples, we point the reviewer to Table 5. Here we use the DAT objective with precomputed DAD samples. We find that training on DAD adversarial examples without distillation can also improve performance on natural distribution shifts but to a smaller extent.
> \
> \
> *The adversarial examples could also not be computed but rather perturbations could be sampled instead (while still using the VQGAN), or random data augmentation methods (or just an identity function used, i.e. which would correspond to just dropping the last l_2 loss in Eq 9)...*
>
> We thank the reviewer for the interesting suggestion. In fact, we already provide results using common data augmentations. This can be observed in Table 7 in the Appendix. Here we use Mixup and RandAugment, which are components of AugReg, and demonstrate the results of distilling from CLIP without using the generative model. Since these augmentations replace the base image and do not add additional samples, the two L_2 losses are combined. The results are much worse, with an average -13.2% decrease in performance on natural distribution shifts. We will provide the results of sampling the VQGAN in the discussion phase.
> \
> \
> *In Table 1, does the row "DAT + DAD (Ours)" refer to training with DAT & DAD on top of AugReg-ViT[55]?*
>
> Yes, that is correct, DAT + DAT is also on top of AugReg. We will update the table with the “+”
> \
> \
> *In line 303, the paper states "These adversarial examples are generated with DAD" -- was "DAT" intended?*
>
> No, the original statement is correct. The goal of this ablation is to see how DAD examples improve over DAT examples.
> \
> \
> We hope our response addressed the reviewer’s questions and concerns. We are happy to answer any further questions.
>
> Jieru Mei, Yucheng Han, Yutong Bai, Yixiao Zhang, Yingwei Li, Xianhang Li, Alan Yuille, and Cihang Xie. Fast advprop. ICLR, 2022

---

> > ### Author Response · Authors · 2023-08-11
> > **Follow up on data augmentation ablation**
> >
> > The reviewer asked about sampling from the VQGAN without computing the gradient. As promised in the initial response we provide an additional result which was not finished then on sampling from the VQGAN without using gradients.
> >
> > The results are significantly worse than DAD, indicating the need to use gradients to discover diverse samples. This is also supported by our theoretical analysis that indicates more diverse adversarial examples are better for robustness. Higher in-distribution performance also suggests the samples are less diverse.
> >
> > | | ImageNet | V2 | R | Sketch | A | Avg |
> > | --- | --- | --- | --- | --- | --- | --- |
> > | VQGAN - Sample | 80.9 | 70.1 | 49.3 | 34.9 | 24.0 | 51.8 |
> > | VQGAN - Grad | 79.6 | 69.9 | 65.1 | 46.1 | 31.8 | 69.5 |
> >
> > We hope this result is helpful and improves our response to the reviewer's question.

---

### Official Review · Reviewer_gGrQ · 2023-07-07

**Soundness:** 3 good
**Presentation:** 2 fair
**Contribution:** 3 good
**Rating:** 5
**Confidence:** 3

**Summary:**

This paper introduces a novel approach called discrete adversarial distillation (DAD) to train small, robust models using large-scale models as teachers. The authors establish the knowledge distillation framework for out-of-distribution robustness and provide a theoretical framework for utilizing large-scale models as teachers. DAD achieves state-of-the-art performance on natural distribution shifts, surpassing adversarial training and traditional distillation techniques.

**Strengths:**

Overall, this paper is also well-written and clearly structured; the authors provide both theoretical analysis and empirical experiments to verify their approach. The reviewer would like to list some strengths of the work.

Firstly, it extends the knowledge distillation framework to the domain of out-of-distribution robustness, opening up new possibilities for training small, robust models that can generalize effectively across diverse populations and environments.

Secondly, this paper highlights the advantages of leveraging large-scale models as teachers, providing a theoretical framework to support this approach. This understanding can inform the development and utilization of robust models in practical applications.

Lastly, the experimental results showcase state-of-the-art performance on natural distribution shifts, surpassing existing techniques. This signifies the potential of the proposed DAD method in addressing the challenges of robustness and generalization in ML models.

**Weaknesses:**

The reviewer found it hard to understand the main idea at first glance, the paper could improve its clarity in figures, as well as in presenting the assumptions and theoretical analysis, and its connection to the proposed method. Explicitly linking the theoretical framework to the practical implementation of the DAD method would help readers grasp the underlying principles and motivations behind the proposed approach more effectively.



**Questions:**

Please refer to the weakness. In addition, why the authors did not provide the code for review but answered the *Reproducibility* as "yes"? Please provide the code during the rebuttal period, using Anonymous GitHub is a viable option. Also, the authors claim that the proposed only brings negligible computational overhead, but the reviewer could not find the part that addresses computing resources; what exactly are the computational costs for the proposed framework?

---

> ### Author Rebuttal · Authors · 2023-08-10
>
> We thank the reviewer for the constructive comments and positive review. We are glad the reviewer agreed with the core contributions of our setting and the importance of small, robust models.
> \
> \
> *The reviewer found it hard to understand the main idea at first glance, the paper could improve its clarity in figures, as well as in presenting the assumptions and theoretical analysis, and its connection to the proposed method. Explicitly linking the theoretical framework to the practical implementation of the DAD method would help readers grasp the underlying principles and motivations behind the proposed approach more effectively.*
>
> 1. We have attached an updated main figure in the PDF.
> 2. In Sec 3.4, our goal is to establish the conditions where robustness is achieved through data augmentation / adversarial training. We find that the diversity of an adversarial sample (measured by Wasserstein distance from the training distribution) depends on the diversity of the model’s training distribution, so we propose using the large-scale teacher’s gradients to generate this. This can be mapped to L163 in Eq. 9 if we substitute α(ϕ(P1)) for DAD and α(ϕ(P2)) for DAT in Lemma 3.4. We will revise the section with this explicit notation.
> 3. For a qualitative result linking our theory and implementation, we kindly point the reviewer to the Appendix, where we provide figures that demonstrate the relationship between the magnitude of the distribution shift to performance. In figure B, we find that contrary to standard ImageNet training and DAT, DAD has improved performance when the Wasserstein distance of the evaluation distribution is higher. This is also shown in figure C, where we show that the improvement over standard training with DAD is higher than with DAT. This reflects our theoretical conclusions that DAD adversarial examples are more informative. We moved this result to the Appendix due to space, but agree with the reviewer that demonstrating this connection is important and will add this to the main paper in the revised version.
>
> \
> *In addition, why the authors did not provide the code for review but answered the Reproducibility as "yes"? Please provide the code during the rebuttal period, using Anonymous GitHub is a viable option*
>
> Thanks for bringing this to our attention, we will attach the code following rebuttal guidelines.
> \
> \
> *Also, the authors claim that the proposed only brings negligible computational overhead, but the reviewer could not find the part that addresses computing resources; what exactly are the computational costs for the proposed framework?*
>
> We apologize for not being clear about this claim. We have attached a table with the results of an empirical analysis following the methodology in FastAdvProp [1] of the full computational cost of DAD. To clarify, what we mean by “negligible computational overhead” is the cost of generating adversarial examples, which is the largest cost of adversarial training. Compared to standard adversarial training and DAT, since we generate our samples from a frozen teacher and reuse them, the cost of generation over a full training run becomes negligible. The only major additional cost over standard training is training on the new examples, which is also shared by other forms of adversarial training. We will update the claim with the precise statement in the revised paper.
>
> | Method               | Attack Steps | Training Budget |
> |----------------------|--------------|-----------------|
> | ImageNet | 0           | 1x             |
> | Adversarial Training | 10           | 11x            |
> | AdvProp              | 5            | 7x              |
> | AdvProp              | 1            | 3x              |
> | DAT                  | 1            | 3.5x            |
> | DAD                  | 1            | 2x              |
>
>
> \
> We hope our response addressed the reviewer’s questions and concerns. We are happy to answer any further questions.
>
> [1] Jieru Mei, Yucheng Han, Yutong Bai, Yixiao Zhang, Yingwei Li, Xianhang Li, Alan Yuille, and Cihang Xie. Fast advprop. *ICLR*, 2022

---

> > ### Comment · Reviewer_gGrQ · 2023-08-15
> >
> > I have carefully read the authors' rebuttal and reviewed the submitted code. I thank the authors for their effort and time in addressing my questions. I have no further comments.

---

### Official Review · Reviewer_GKv1 · 2023-07-08

**Soundness:** 3 good
**Presentation:** 3 good
**Contribution:** 2 fair
**Rating:** 5
**Confidence:** 4

**Summary:**

This paper presents a knowledge distillation framework for vision model, by leveraging a teacher model (CLIP). In the setup, a discretizer (VQGAN) model is introduced to the adversarial examples from the teacher model. Adversarial training (AT) objective is adapted in the knowledge distill setting. The proposed discrete adversarial distillation (DAD) demonstrate improved performance in out-of-distribution tasks, comparing to other AT approaches.

**Strengths:**

This paper present an interesting approach for knowledge distillation for large scale vision model. The experiment is thoroughly conducted with various of SOTA approaches compared, and comprehensive ablation study. The writing is easy to follow and the idea is well established.

**Weaknesses:**

1. One main concern is on comparing the proposed method with DAT. In DAT, the idea of discretizer is introduced and VQGAN was also considered in their framework. In their paper, various tasks were evaluated, including image classification, self-supervised learning, and object detection. To me, DAD seems to be applying the similar methodology, but on a OOD setup.

2. In the experiment, it seems like DAT has achieved competitive results in ImageNet-1K, and even better results in ImageNet-21K. What is the key advantage of DAD over DAT?

**Questions:**

1. How come DAD+DAT performs worse than DAT in some cases?
2. Are there any qualitative results? what are the drawbacks from previous model(s) that are better addressed by the newly proposed DAD?
3. Have you try training the teacher model in parallel with the student model, not just provide adversarial samples from offline manner?

**Limitations:**

Please see weakness.

---

> ### Author Rebuttal · Authors · 2023-08-10
>
> We thank the reviewer for the insightful comments and positive review. We are glad the reviewer found our idea well established, experiments thorough and comprehensive, and writing easy to follow. We address the concerns about the comparisons and results below.
> \
> \
> *One main concern is on comparing the proposed method with DAT...To me, DAD seems to be applying the similar methodology, but on a OOD setup.*
>
> Although DAD draws upon the use of a VQGAN for data augmentation, the methodology is distinct in three key ways.
> 1. *Setting.* DAT follows a standard data augmentation setting to improve robustness. We are the first to also introduce a robust teacher and use knowledge distillation to further improve robustness. We focus on OOD-robustness due to practical usefulness and the specialization of large-scale models on natural distribution shifts. These models exhibit strong robustness due to their diverse training data, and we are the first to leverage these representations as an additional form of regularization
> 2. *Objective.* We introduce a novel knowledge distillation objective for the OOD-robustness setting where we add a second KL-divergence term between the student and teacher predictions on the augmented image. Previous works in defensive distillation use the teacher predictions on the normal image for this second term, but this is not adaptive to semantic transformations.
> 3. *Data augmentation.* This is the aspect of DAD that is adapted from DAT. However, our key novelty here is using the teacher’s gradients to generate adversarial examples. We find that this results in more diverse adversarial examples closer to the teacher’s representations, and our theoretical framework is based on this idea. This also makes DAD much cheaper than DAT since we only generate the adversarial examples once.
>
> \
> *In the experiment, it seems like DAT has achieved competitive results in ImageNet-1K, and even better results in ImageNet-21K. What is the key advantage of DAD over DAT?*
>
> Although DAT performs well, we observe distinct improvements in DAD across distributions and especially on natural distribution shifts.
>
> - On average, there is an improvement of 2.7% for ViT-B and 4.2% for ResNet50 on ImageNet-1K across 7 distribution shifts.
> - Also following Reviewer Y3Hr’s suggestion, we updated Table 2 with the average performance for clarity. *DAD has higher results than DAT on ImageNet-21K as well.* We attached the updated table in the uploaded PDF.
> - DAD has a significant improvement of (avg +10.3% for ViT-B and avg +7.1% for ResNet50) on natural distribution shifts  (IM-A, IM-R, IM-Sketch). The key advantage of DAD over DAT is that it is able to leverage the robustness of a pretrained foundation model. This is also the likely cause for worse performance on synthetic distribution shifts. In fact, the CLIP teacher performs worse than DAT training on ImageNet-C (-15.5%) and ImageNet-Stylized (-4.6%) due to being trained only on natural images, making it difficult to surpass DAT on these distributions with a distillation-based approach.
>
> \
> *How come DAD+DAT performs worse than DAT in some cases?*
>
> We found that combining DAD with DAT resulted in higher in-distribution accuracy and accuracy on synthetic distribution shifts, but at the cost of lower performance on natural distribution shifts than DAD. In practice, DAD targets natural distribution shifts while DAT performs better on synthetic corruptions. Due to CLIP’s diverse training set, our theoretical framework in Sec 3.4 shows CLIP’s adversarial examples encode similar representations. Combining them may dilute the specialization that leads to their strong individual performance. We do note that DAD+DAT still outperforms DAT on natural distribution shifts.
>
> \
> *Are there any qualitative results? what are the drawbacks from previous model(s) that are better addressed by the newly proposed DAD?*
>
> Yes we included qualitative results:
> 1. We add visualizations of the generated images for DAD in the uploaded PDF. Both DAT and DAD adversarial examples are computed with the same budget. We select sets where the DAT image is classified correctly but the DAD image is classified incorrectly to highlight the difference. DAD images appear to be more graphic, displaying a greater variety of distribution shifts than the corresponding DAT image.
> 2. We kindly point the reviewer to the Appendix where we provide graphs in Figures A-C on page 15 describing the connection to our theoretical framework. Here we demonstrate the relationship between Wasserstein distance of the evaluation distribution and performance. In figure C, we show that for a test distribution, DAD has a larger improvement over standard ImageNet training than DAT the larger the Wasserstein distance between the test distribution and training distribution is. This suggests DAD can better address harder distribution shifts like natural distribution shifts (IM-A, IM-R, IM-Sketch), which are also closer real-world use cases than simple synthetic modifications to the base image (IM-C, IM-Style).
> 3. DAD is also cheaper than DAT and AT. We find that DAT is only 2x as expensive as standard training, compared to 3.5x for DAT and 11x for AT.
>
> \
> *Have you try training the teacher model in parallel with the student model, not just provide adversarial samples from offline manner?*
>
> This is an interesting suggestion and something we started to try before deciding not to pursue further. Joint training certainly has the potential to discover harder and more diverse examples, but at a higher computational cost of updating two models. In addition, it is still unclear what the best way to apply adversarial training to CLIP is, especially on only ImageNet data. We found that training the teacher led to instability, but this is definitely an idea that could lead to further improvements.
>
> \
> We hope our response addressed the reviewer’s questions and concerns. We are happy to answer any further questions.

---

### Author Rebuttal · Authors · 2023-08-10

We are thankful for the generally positive reviews and useful feedback. We are glad reviewers found our idea novel (Reviewers gGrQ, Y3Hr, qyW3), setting significant (Reviewers gGrQ, Y3HR), and analysis convincing (Reviewers GKv1, qyW3, ktZ5). We provide an updated main figure, updated Table 2, and additional visualizations of DAD adversarial examples. To address specific concerns, we also provide additional experimental results on using Stable Diffusion, computational cost, and adversarial robustness.

---

### Decision · Program_Chairs · 2023-09-21

**Decision:**

Accept (poster)

**Comment:**

The recommendation is based on the reviewers' comments, the area chair's personal evaluation, and the post-rebuttal discussion.

This paper studies a new way of generating discrete adversarial examples with the aid of CLIP models. All reviewers find the studied setting novel and the results provide new insights. The authors’ rebuttal has successfully addressed the major concerns of reviewers, and the reviewers' comments and suggestions have helped sharpen the results and presentations. All reviewers are in favor of accepting this submission. Therefore, I recommend acceptance of this submission. I also expect the authors to include the new results and suggested changes during the rebuttal phase to the final version.